# Initialization Schemes for Kolmogorov–Arnold Networks: An Empirical Study

**Spyros Rigas** *
Department of Digital Industry Technologies
National and Kapodistrian University of Athens
344 00 Psachna, Greece
`spyrigas@uoa.gr`

**Dhruv Verma**
Applied and Computational Mathematics
California Institute of Technology
Pasadena, CA 91125, United States of America
`dverma@caltech.edu`

**Georgios Alexandridis**
Department of Digital Industry Technologies
National and Kapodistrian University of Athens
344 00 Psachna, Greece
`gealexandri@uoa.gr`

**Yixuan Wang**
Applied and Computational Mathematics
California Institute of Technology
Pasadena, CA 91125, United States of America
`roywang@caltech.edu`

## Abstract

Kolmogorov–Arnold Networks (KANs) are a recently introduced neural architecture that replace fixed nonlinearities with trainable activation functions, offering enhanced flexibility and interpretability. While KANs have been applied successfully across scientific and machine learning tasks, their initialization strategies remain largely unexplored. In this work, we study initialization schemes for spline-based KANs, proposing two theory-driven approaches inspired by LeCun and Glorot, as well as an empirical power-law family with tunable exponents. Our evaluation combines large-scale grid searches on function fitting and forward PDE benchmarks, an analysis of training dynamics through the lens of the Neural Tangent Kernel, and evaluations on a subset of the Feynman dataset. Our findings indicate that the Glorot-inspired initialization significantly outperforms the baseline in parameter-rich models, while power-law initialization achieves the strongest performance overall, both across tasks and for architectures of varying size. This work underscores initialization as a key factor in KAN performance and introduces practical strategies to improve it.

## 1 Introduction

Kolmogorov–Arnold Networks (KANs) (Liu et al., 2025) have recently emerged as an alternative backbone architecture to Multilayer Perceptrons (MLPs), drawing inspiration from the Kolmogorov–Arnold representation theorem (Kolmogorov, 1957) in a manner analogous to how the learning of MLPs relies on universal approximation theorems. Unlike MLPs, which use fixed nonlinear activation functions and trainable synaptic weights, KANs comprise grid-dependent trainable activation functions. This provides them with flexibility in modeling complex nonlinear relationships, while requiring fewer and smaller layers. Since their introduction, KANs have found numerous applications, often surpassing the performance of their MLP-based counterparts (Yu et al., 2024; Poeta et al., 2024). There have been many notable results in scientific problem-solving domains, including function fitting and symbolic regression (Liu et al., 2024; Shukla et al., 2024), partial differential equations (PDEs) (Shukla et al., 2024; Rigas et al., 2024; Wang et al., 2025b) and operator learning (Abueidda et al., 2025; Shukla et al., 2024; Lee et al., 2025), among other applications (Howard et al., 2024; Kundu et al., 2024; Kashefi, 2025).

Beyond these benchmarks, there has also been significant progress in the theoretical understanding of KANs (Zhang & Zhou, 2025; Alter et al., 2025; Wang et al., 2025a). However, one important theoretical and practical aspect that remains understudied pertains to their initialization strategies.

---

*Corresponding author.

Current literature mainly relies on the standard initialization method proposed in the introductory KAN paper (Liu et al., 2025), or explores alternative KAN variants such as Chebyshev-based formulations (Rigas et al., 2026). This highlights a clear gap and motivates an investigation into more effective initialization approaches for the standard spline-based architecture. Effective initialization is crucial, as a good "initial guess" for the network weights can significantly accelerate training (Mishkin & Matas, 2016; Skorski et al., 2021) and prevent early saturation of hidden layers (Glorot & Bengio, 2010). However, despite extensive research into initialization methods for MLP-based architectures, these results cannot be directly applied to KANs. Furthermore, even within MLP-based architectures, initialization methods often require separate consideration depending on the specific architecture design (Huang et al., 2020), activation function (He et al., 2015), or even on a complete case-by-case basis (Skorski et al., 2021).

In response to this research gap, this work explores initialization strategies for the standard, spline-based KAN architecture. Drawing parallels with MLPs, we propose variance-preserving schemes inspired by LeCun (LeCun et al., 1998) and Glorot (Glorot & Bengio, 2010) initializations, including a variant that employs normalized spline basis functions. In addition, recognizing that theoretical frameworks may not always align with empirical performance (Mishkin & Matas, 2016), we further propose an empirical family of power-law initializations parameterized by two exponents. We perform grid searches to identify suitable exponent choices for the power-law method on function fitting and forward PDE benchmarks and then evaluate all initialization schemes on these tasks. We subsequently fix the exponents to values that lie within the identified well-performing range and select two representative architectures (in terms of parameter count) to analyze training dynamics through the evolution of training loss curves and the Neural Tangent Kernel (NTK) spectrum (Jacot et al., 2018; Wang et al., 2022). Finally, we evaluate said architectures on a subset of the Feynman dataset (Udrescu & Tegmark, 2020), which, although widely used for symbolic regression, is formulated here as a function fitting benchmark as in Liu et al. (2025).

## 2 BACKGROUND

### 2.1 KOLMOGOROV–ARNOLD NETWORKS

Within the standard formalism, the output, $\mathbf{y} \in \mathbb{R}^{n_{\text{out}}}$, of a KAN layer is related to its input, $\mathbf{x} \in \mathbb{R}^{n_{\text{in}}}$, via:

$$y_j = \sum_{i=1}^{n_{\text{in}}} \left( r_{ji}\, R\,(x_i) + c_{ji} \sum_{m=1}^{G+k} b_{jim}\, B_m\,(x_i) \right), \quad j = 1, \ldots, n_{\text{out}}, \tag{1}$$

where $r_{ji}$, $c_{ji}$ and $b_{jim}$ are the layer's trainable parameters, $R(x)$ corresponds to a residual function, typically chosen as the SiLU, i.e., $R(x) = x\,(1 + e^{-x})^{-1}$, and $B_m\,(x)$ denotes a univariate spline basis function of order $k$, defined on a grid with $G$ intervals. For each of the layer's trainable parameters, the original KAN formulation initializes the scaling weights as $c_{ji} = 1$, the residual weights $r_{ji}$ using Glorot initialization (Glorot & Bengio, 2010), and the basis weights $b_{jim}$ from a normal distribution with zero mean and small standard deviation, typically set to $\sigma = 0.1$. We will henceforth refer to this configuration as the "baseline initialization".

### 2.2 RELATED WORK

In the existing KAN literature, initialization strategies have only been explored in certain KAN variants (see, e.g., Guilhoto & Perdikaris (2025)), while the standard spline-based architecture has not yet received dedicated attention in this regard. A natural starting point for studying initialization is to follow the historical developments in MLP-based architectures, beginning with variance-preserving schemes such as those proposed by LeCun (LeCun et al., 1998) and Glorot (Glorot & Bengio, 2010), which stabilize activation variance across layers and mitigate progressive vanishing or explosion. Within the KAN family, Glorot-inspired initialization has been applied successfully to Chebyshev-based variants (Rigas et al., 2026), though this setting differs substantially from the spline-based case studied here, since it removes the residual term of Eq. (1) and employs a completely different basis function. Consequently, it remains unclear whether such strategies directly transfer to the standard

KAN formulation, motivating the investigation presented in this work. To the best of our knowledge, the present work provides the first systematic study of initialization strategies for spline-based KANs.

# 3 METHODOLOGY

## 3.1 PROPOSED INITIALIZATIONS

Since the three weight types in a KAN layer are independent, we may initialize the scaling weights $c_{ji}$ to 1 and focus exclusively on the initialization of the residual weights $r_{ji}$ and basis weights $b_{jim}$. We assume that these weights are drawn from zero-mean distributions with standard deviations $\sigma_r$ and $\sigma_b$, respectively. To determine suitable values for $\sigma_r$ and $\sigma_b$, we follow the principle of variance preservation proposed by LeCun (LeCun et al., 1998), which stipulates that the variance of the outputs of each layer should match that of its inputs, thereby avoiding amplification or attenuation of the signal across layers. Assuming statistical independence among terms and an equal contribution to the variance from each of the $(G+k+1)$ terms in the summand of Eq. (1), we derive the following expressions for the standard deviations[1]:

$$\sigma_r = \sqrt{\frac{\text{Var}(x_i)}{n_{\text{in}}(G+k+1)\,\mu_R^{(0)}}}, \qquad \sigma_b = \sqrt{\frac{\text{Var}(x_i)}{n_{\text{in}}(G+k+1)\,\mu_B^{(0)}}}, \tag{2}$$

where

$$\mu_R^{(0)} = \mathbb{E}\left[R\left(x_i\right)^2\right], \qquad \mu_B^{(0)} = \mathbb{E}\left[B_m\left(x_i\right)^2\right], \tag{3}$$

with $\mu_B^{(0)}$ denoting the expectation over both the input distribution and all spline basis indices, $m$, and $\mu_R^{(0)}$ denoting the expectation over the input distribution alone.

If we further assume that each component of $\mathbf{x}$ is drawn from a given distribution (e.g., the uniform distribution $\mathcal{U}\left(-1, 1\right)$, as is often the case in tasks like function fitting or PDE solving), then all statistical quantities in Eq. (2) can be evaluated directly, except for $\mu_B^{(0)}$. Due to the dependence of the spline-basis functions on the underlying grid, no general analytic expression exists for $\sigma_b$. This leads to two practical alternatives: one may either estimate $\mu_B^{(0)}$ numerically by sampling a large number of input points from the assumed distribution at initialization, or set the expectation value to unity by modifying the architecture of the KAN layer to use normalized spline basis functions, defined as

$$\tilde{B}_m\left(x_i\right) = \frac{B_m\left(x_i\right) - \mathbb{E}\left[B_m\left(x_i\right)\right]}{\sqrt{\mu_B^{(0)} - \mathbb{E}^2\left[B_m\left(x_i\right)\right]}}, \tag{4}$$

where the expectation values are computed over the layer inputs during the forward pass. We will refer to the former alternative as "LeCun–numerical" initialization, while the latter is referred to as "LeCun–normalized" initialization.

While these LeCun-inspired schemes focus on preserving the variance of forward activations, they do not explicitly account for the propagation of gradients. To address this, we also consider a Glorot-inspired initialization, which aims to balance forward- and backward-pass variance by maintaining stable variance for both activations and gradients across layers. Under the same assumptions as before, we derive the following expressions for the standard deviations[2]:

$$\sigma_r = \sqrt{\frac{1}{G+k+1} \cdot \frac{2}{n_{\text{in}}\,\mu_R^{(0)} + n_{\text{out}}\,\mu_R^{(1)}}}, \qquad \sigma_b = \sqrt{\frac{1}{G+k+1} \cdot \frac{2}{n_{\text{in}}\,\mu_B^{(0)} + n_{\text{out}}\,\mu_B^{(1)}}}, \tag{5}$$

---

[1]See Appendix A for detailed derivations.
[2]See Appendix B for detailed derivations.

where

$$\mu_R^{(1)} = \mathbb{E}\left[R'\left(x_i\right)^2\right], \qquad \mu_B^{(1)} = \mathbb{E}\left[B_m'\left(x_i\right)^2\right], \tag{6}$$

with the expectations defined analogously to $\mu_R^{(0)}$ and $\mu_B^{(0)}$ in Eq. (3). In practice, $\mu_B^{(1)}$ can be computed using automatic differentiation of the spline basis functions, together with the numerical sampling strategy discussed for the LeCun–numerical case, while $\mu_R^{(1)}$ can be evaluated analytically for standard choices of $R(x)$ such as the SiLU.

In addition to these theory-driven initialization strategies, we also investigate an empirical approach based on a power-law scaling of the KAN layer's architectural parameters. Specifically, we initialize the weights such that their standard deviations follow the form

$$\sigma_r = \left(\frac{1}{n_{\text{in}}\left(G + k + 1\right)}\right)^{\alpha}, \qquad \sigma_b = \left(\frac{1}{n_{\text{in}}\left(G + k + 1\right)}\right)^{\beta}, \tag{7}$$

where $\alpha$ and $\beta$ are tunable exponents selected from the set $\{0.0, 0.25, \dots, 1.75, 2.0\}$. The motivation behind this empirical scheme is to perform a grid search over $(\alpha, \beta)$ configurations in order to identify trends or specific exponent pairs that consistently improve training speed and convergence. Such searches can be carried out on a per-domain basis (e.g., function fitting, forward PDEs), after which the resulting well-performing ranges may serve as reusable heuristics for future problems of the same type.

## 3.2 Experimental Setup

We evaluate initialization strategies on two benchmark families: function fitting tasks and forward PDE problems. For function fitting, we use five two-dimensional target functions and train for 2,000 iterations (epochs), while for PDEs we consider the Allen–Cahn equation, Burgers' equation, and the two-dimensional Helmholtz equation, using KANs trained for 5,000 iterations within the Physics-Informed Machine Learning (PIML) framework (Raissi et al., 2019). Across both benchmarks, performance is measured using the final training loss and the relative $L^2$ error with respect to the reference solution. For the purposes of the initial grid search, we test architectures with 1–4 hidden layers, widths equal to $2^i$ for $i = 1, \dots, 6$, and grid sizes $G \in \{5, 10, 20, 40\}$ for function fitting, while for PDEs we restrict to $G \in \{5, 10, 20\}$. All experiments presented herein are repeated with five random seeds, except in the power-law grid search where we use three seeds to reduce computational cost, and we report the median outcome across runs. Further implementation details, including the explicit formulas of the target functions and the PDE setups, are provided in Appendix C. All experiments are implemented in JAX (Bradbury et al., 2018), with KANs trained using the jaxKAN framework (Rigas & Papachristou, 2025). Training is performed on a single NVIDIA GeForce RTX 4090 GPU.

## 4 Experiments & Discussion

### 4.1 Grid-Search Results

The grid search over $(\alpha, \beta)$ configurations and the architectural variations described in Section 3.2 resulted in 126,240 trained KAN model instances for function fitting. After aggregating the repeated runs by their median outcome, this number reduces to 40,800 representative results. From these, we retain only the best-performing $(\alpha, \beta)$ configuration per setting, yielding 2,400 final entries. Table 1 reports, for each target function and initialization scheme, the percentage of runs that outperform the baseline initialization described in Section 2.1. Results are compared with respect to final training loss and relative $L^2$ error, and we additionally report the percentage of runs where both metrics improve simultaneously.

The LeCun-inspired schemes rarely outperform the baseline on the smaller architectures, but their effectiveness increases with depth, width and grid size. In some of the larger settings, the normalized variant in particular achieves improvements exceeding two orders of magnitude relative to

the baseline. However, in terms of absolute frequency, Table 1 clearly shows that the baseline still outperforms both LeCun-based variants: for more than 70% of configurations, the resulting relative $L^2$ error under LeCun initialization is higher than under the baseline scheme. Between the two LeCun variants, the normalized version consistently performs better than the numerical one, which is consistent with its design, as variance preservation is enforced by construction. On the other hand, the Glorot-inspired initialization performs more robustly. Across all five functions, it yields success rates of approximately 60–75% for both loss and relative $L^2$ error, indicating that simultaneously balancing forward- and backward-pass variances is considerably more effective than forward-variance preservation alone. The few cases where the baseline performs better occur predominantly for the smaller architectures.

As far as the power-law initialization is concerned, it exhibits the strongest overall performance. Table 1 shows that, for nearly every architecture and target function, there exists at least one $(\alpha, \beta)$ pair that outperforms the baseline, often by a substantial margin. The most favorable region is concentrated around small residual exponents, i.e., $\alpha \in \{0.25, 0.5\}$, combined with larger basis-function exponents, namely $\beta \geq 1.0$. Full grid-search results illustrating these trends are provided in Appendix D and the supplementary material. Notably, even when fixing a single configuration within this region, the method remains highly robust. For example, with $(\alpha, \beta) = (0.25, 1.0)$, the initialization yields simultaneous improvements in both loss and $L^2$ error over the baseline in 96.88% of runs for $f_1(x, y)$, 95.83% for $f_2(x, y)$, 97.92% for $f_3(x, y)$, 87.50% for $f_4(x, y)$, and 89.58% for $f_5(x, y)$. This indicates that once a suitable exponent range is identified for a given problem type, a fixed choice within that region can systematically outperform both the baseline and the Glorot initialization.

Following the same procedure for the PDE benchmarks, we trained 56,882 models, a number which reduced to 18,360 representative results after aggregation and 1,080 final entries after selecting the best $(\alpha, \beta)$ per setting. Table 2 summarizes the outcomes in terms of final training loss, relative $L^2$ error, and their joint improvement over the baseline initialization.

When comparing the two LeCun-based schemes to each other, the observed behavior is essentially the inverse of what we observed in function fitting. The normalized variant, which performed better than the numerical one for function fitting, fails almost entirely in PDE problems: in nearly all configurations, it produces no improvement over the baseline, with success rates effectively equal to zero. This discrepancy can be attributed to the fact that PDE losses involve not only the network output but also its derivatives and nonlinear combinations thereof. Normalizing the spline basis propagates all multiplicative constants (the standard deviation in the denominator of Eq. (4) in this case) into all derivatives, altering the stiffness of the residuals. While the numerical variant avoids this issue, the results are still poor: although it occasionally outperforms the baseline on larger architectures, it remains generally ineffective. The Glorot-inspired initialization again shows more consistent improvements. As in function fitting, it performs significantly better than the baseline on parameter-rich architectures and the cases where it underperforms correspond to smaller models.

Table 1: Percentage of runs that outperform the baseline initialization on function fitting benchmarks. Columns correspond to target functions, while rows correspond to initialization schemes and evaluation metrics. Best results per function are shown in bold.

| Initialization | Metric | $f_1(x, y)$ | $f_2(x, y)$ | $f_3(x, y)$ | $f_4(x, y)$ | $f_5(x, y)$ |
|---|---|---|---|---|---|---|
| LeCun–numerical | Loss | 18.75% | 14.58% | 12.50% | 25.00% | 26.04% |
| | $L^2$ | 6.25% | 4.17% | 5.21% | 14.58% | 2.08% |
| | Both | 1.04% | 0.00% | 0.00% | 8.33% | 0.00% |
| LeCun–normalized | Loss | 19.79% | 28.13% | 19.79% | 41.67% | 31.25% |
| | $L^2$ | 11.46% | 9.38% | 11.46% | 26.04% | 6.25% |
| | Both | 2.08% | 5.21% | 5.21% | 16.67% | 1.04% |
| Glorot | Loss | 78.13% | 76.04% | 78.13% | 63.54% | 72.92% |
| | $L^2$ | 78.13% | 75.00% | 78.13% | 64.58% | 72.92% |
| | Both | 78.13% | 75.00% | 78.13% | 60.41% | 64.59% |
| Power-Law | Loss | **100.00%** | **100.00%** | **100.00%** | **100.00%** | **98.96%** |
| | $L^2$ | **100.00%** | **100.00%** | **100.00%** | **94.79%** | **96.88%** |
| | Both | **100.00%** | **100.00%** | **100.00%** | **94.79%** | **95.83%** |

Table 2: Percentage of runs that outperform the baseline initialization on forward PDE benchmarks. Columns correspond to the three PDEs considered, while rows correspond to initialization schemes and evaluation metrics. Best results per PDE are shown in bold.

| Initialization | Metric | Allen–Cahn | Burgers | Helmholtz |
|---|---|---|---|---|
| LeCun–numerical | Loss | 11.11% | 11.11% | 8.33% |
| | $L^2$ | 16.67% | 22.22% | 15.28% |
| | Both | 8.33% | 6.94% | 2.78% |
| LeCun–normalized | Loss | 2.78% | 0.00% | 0.00% |
| | $L^2$ | 0.00% | 0.00% | 0.00% |
| | Both | 0.00% | 0.00% | 0.00% |
| Glorot | Loss | 55.56% | 50.00% | 76.39% |
| | $L^2$ | 51.39% | 54.17% | 72.22% |
| | Both | 41.67% | 36.11% | 62.50% |
| Power-Law | Loss | **98.61%** | **100.00%** | **98.61%** |
| | $L^2$ | **94.44%** | **73.61%** | **87.50%** |
| | Both | **94.44%** | **73.61%** | **87.50%** |

The power-law initialization remains the strongest of all approaches, though the advantage is less pronounced in the PDE case. For more than 70% of configurations (up to 94.44% in the Allen–Cahn equation) there exists at least one $(\alpha, \beta)$ pair that outperforms the baseline simultaneously in both loss and relative $L^2$ error. The preferred exponent region differs slightly from the function fitting case: while small residual exponents $\alpha$ remain strongly favored, in this case lower values of $\beta$ (typically $0.75 \leq \beta \leq 1.25$) yield the best results. The complete grid-search results supporting these observations are provided in Appendix D and the supplementary material. For the previously discussed configuration $(\alpha, \beta) = (0.25, 1.0)$, the power-law initialization outperforms the baseline on both metrics in 83.33% of runs for Allen–Cahn, 54.93% for Burgers' and 59.72% for Helmholtz, with even higher success rates when each metric is considered individually.

## 4.2 TRAINING DYNAMICS ANALYSIS

The previous experiments established that the Glorot- and power-law-based schemes provide strong alternatives to the baseline initialization, with the latter yielding the most consistent improvements overall. In contrast, the LeCun-based variants exhibited substantially weaker performance. To better understand the mechanisms behind these trends, we next examine training dynamics in greater detail for each initialization scheme, excluding only the LeCun-normalized variant due to its complete breakdown in PDE problems for the aforementioned reasons. We begin with the function fitting benchmarks: Figure 1 shows the evolution of the training loss for two representative settings, a "small" architecture ($G = 5$, two hidden layers with 8 neurons each) and a "large" architecture ($G = 20$, three hidden layers with 32 neurons each). For consistency across experiments, we fix the power-law parameters to $\alpha = 0.25$ and $\beta = 1.75$, which lie within the range identified as favorable in the grid search for function fitting. The curves are averaged over five seeds, with shaded regions indicating the standard error.

Across all settings, the loss curves in Figure 1 reinforce the conclusions drawn from the previous experiments: the power-law initialization consistently outperforms all other schemes, converging both faster and to lower final losses. For the small architecture, the baseline and Glorot initializations behave similarly, whereas for the larger architecture Glorot has a clear advantage over the baseline. The LeCun initialization, while stable and not prone to divergence, persistently underperforms the remaining schemes in both architectures. The oscillatory behavior observed for Glorot and power-law in the large-architecture setting is a consequence of using a fixed learning rate throughout training. We intentionally avoided learning-rate scheduling because initialization and learning-rate adaptability are known to interact in subtle ways (e.g., (Yang et al., 2021)), and our goal was to isolate the effect of initialization alone. In Appendix E we provide results for the same experiments using a learning-rate scheduler; the resulting curves are significantly smoother, however the relative performance ranking of the initialization schemes remains unchanged.

The corresponding analysis for the PDE benchmarks is shown in Figure 2, where again we use the power-law parameters $\alpha = 0.25$ and $\beta = 1.75$, despite the grid search identifying a different optimal region for $\beta$, in order to demonstrate that the method remains robust even when not tuned specifically for PDEs. For the small architecture, all initialization schemes ultimately perform comparably, al-

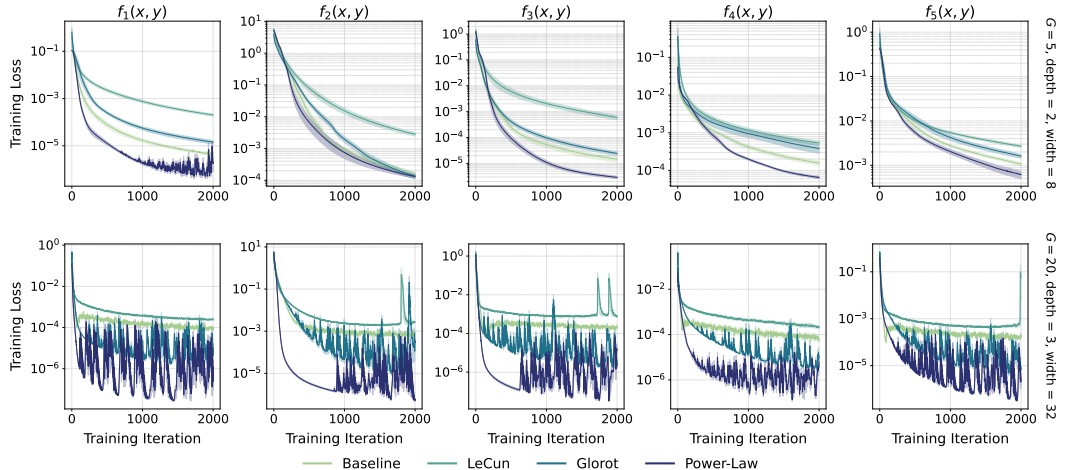

Figure 1: Training loss curves for function fitting benchmarks under baseline, LeCun-numerical, Glorot and power-law ($\alpha = 0.25, \beta = 1.75$) initializations. Results are averaged over five seeds, with shaded regions indicating the standard error. Top row: "small" architecture ($G = 5$, two hidden layers with 8 neurons each). Bottom row: "large" architecture ($G = 20$, three hidden layers with 32 neurons each).

beit at different rates. Notably, the power-law initialization exhibits markedly lower variance across seeds and, in the Allen–Cahn case in particular, reaches a minimum significantly faster than the alternatives. In the larger architecture, however, the trends closely mirror those observed in the function fitting experiments: the LeCun initialization lags behind by several orders of magnitude, the baseline also underperforms, and both Glorot and power-law provide substantial improvements. These two consistently achieve the lowest losses, with the power-law scheme displaying an advantage in the Allen–Cahn and Burgers equations. As with the function fitting results, the oscillatory behavior observed for Glorot and power-law arises from the fixed learning rate used during training. The corresponding smoothed curves obtained with learning-rate scheduling are provided in Appendix E.

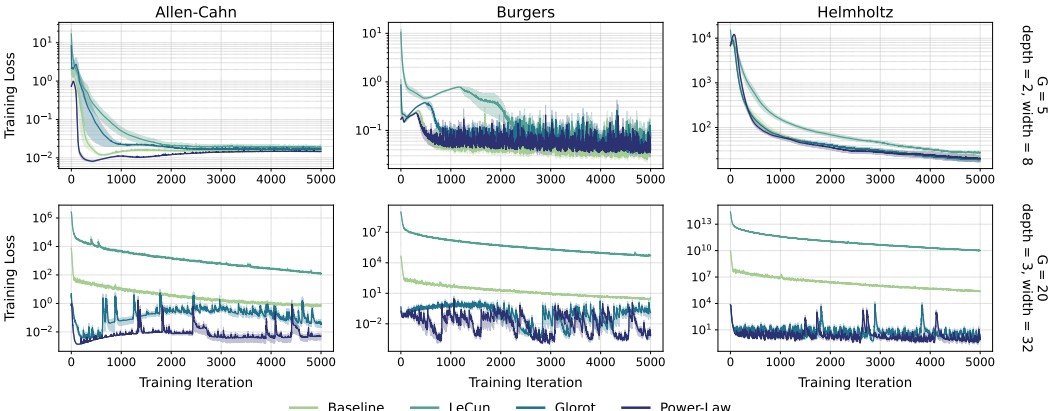

Figure 2: Training loss curves for forward PDE benchmarks under baseline, LeCun-numerical, Glorot, and power-law ($\alpha = 0.25, \beta = 1.75$) initializations. Results are averaged over five seeds, with shaded regions indicating the standard error. Top row: "small" architecture ($G = 5$, two hidden layers with 8 neurons each). Bottom row: "large" architecture ($G = 20$, three hidden layers with 32 neurons each).

To gain further insight into why the baseline and LeCun initializations underperform while Glorot and power-law consistently succeed, we complement the loss-curve analysis with a study of NTK dynamics. Since the discrepancies between initialization strategies are most pronounced in larger

models, we focus here on the "large" architecture. For the power-law method we keep $\alpha = 0.25$ and $\beta = 1.75$, however we also provide the spectra for other configurations in Appendix F.2. Figure 3 shows the NTK eigenvalue spectra at initialization, at intermediate training iterations, and at convergence, for the task of fitting $f_3(x, y)$ (results for other targets are provided in Appendix F.3).

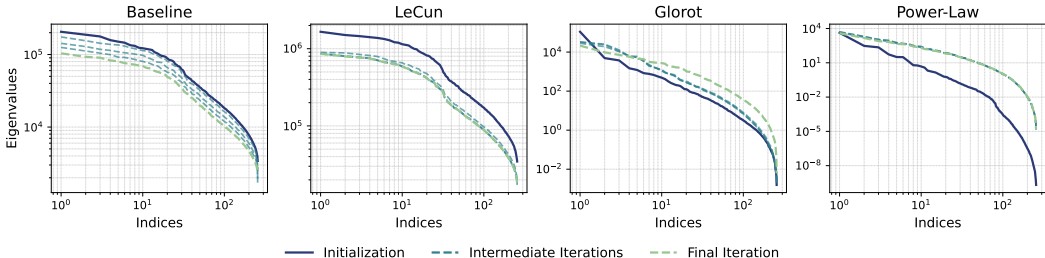

Figure 3: Eigenvalue spectra of the NTK matrix at initialization (solid blue), intermediate iterations (dashed teal), and final iteration (dashed green) for function fitting benchmark $f_3(x, y)$ under different initialization strategies. Results correspond to the "large" architecture ($G = 20$, three hidden layers with 32 neurons each). The power-law initialization uses $\alpha = 0.25, \beta = 1.75$.

The spectra reveal several notable differences. The baseline initialization exhibits a spectrum dominated by very large leading eigenvalues and a steep decay, which collapses further during training, indicating poor conditioning and a low effective rank. LeCun behaves similarly but with even more extreme magnitudes. In contrast, the Glorot initialization produces a well-spread spectrum with stable leading and trailing eigenvalues throughout optimization. The power-law scheme yields an even better spectrum, closely following a power-law decay at initialization and remaining perfectly stable during training, suggesting balanced sensitivity across modes. These observations align with the performance trends reported so far: Glorot and power-law initializations induce stable, well-conditioned NTK spectra and therefore correspond to faster optimization and lower final error, while the baseline and LeCun produce highly skewed or collapsing spectra and thus consistently underperform during training.

We also extend the NTK analysis to PDE benchmarks, focusing on the Allen–Cahn equation as a representative case (results for Burgers and Helmholtz are provided in Appendix F.3). To this end, we adopt the NTK formalism developed for PIML (Wang et al., 2022) and adapt it to account for Residual-Based Attention (RBA) weights (Anagnostopoulos et al., 2024), which are applied in the loss functions studied herein (see Appendix C.2 for details). The resulting kernel is identical to the standard PINN NTK, except that it incorporates the corresponding RBA weights (see Appendix F.1 for the full derivation). Figure 4 shows the NTK eigenvalue spectra separately for the PDE residual term (top row) and the boundary/initial conditions (bottom row).

The PDE residual spectra largely mirror the function fitting case: baseline and LeCun initializations yield poorly conditioned eigenvalues that collapse over training, while both Glorot and power-law maintain stability. The key difference lies in the boundary/initial condition terms, where Glorot shows some irregularities, though far less severe than the baseline and LeCun. The power-law scheme stands out as the most consistent, providing stable and well-structured spectra across both PDE residual- and boundary/initial-term.

### 4.3 FEYNMAN DATASET BENCHMARKS

As a final benchmark, we turn to a subset of the Feynman dataset, restricted to dimensionless equations. The explicit formulas of the target functions and implementation details are provided in Appendix C.3. Having already established via empirical evidence and NTK analysis that the LeCun-based schemes are consistently underperforming, we benchmark here only the two competitive initializations (Glorot and power-law) alongside the baseline. We again fix the power-law exponents to $(\alpha, \beta) = (0.25, 1.75)$ and evaluate all methods using the same "small" and "large" architectures defined in Section 4.2. Tables 3 and 4 report the results in terms of final training loss and relative $L^2$ error with respect to the reference solutions, for the small and large settings, respectively.

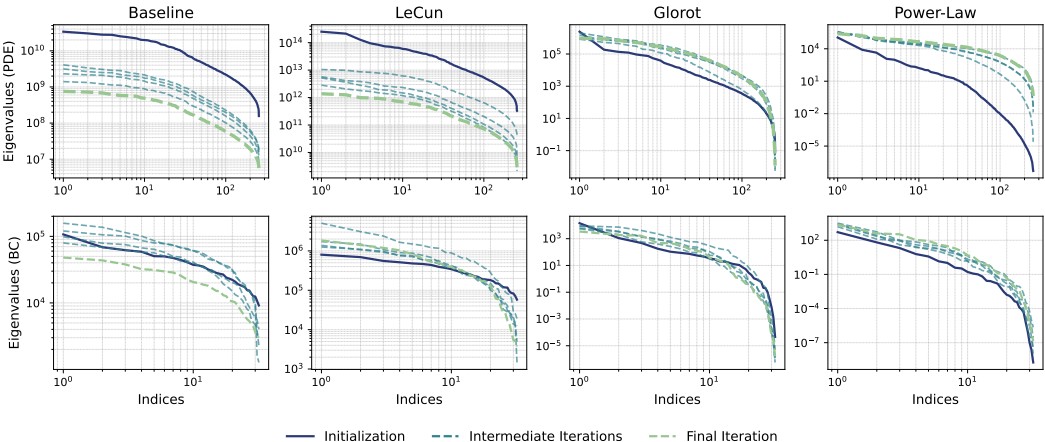

Figure 4: NTK eigenvalue spectra for the Allen–Cahn PDE benchmark under baseline, LeCun-numerical, Glorot, and power-law ($\alpha = 0.25, \beta = 1.75$) initializations. Top row: spectra corresponding to the PDE residual term. Bottom row: spectra for the boundary/initial condition terms. Solid blue lines show the initialization, dashed teal lines show intermediate iterations, and dashed green lines show the final iteration. Results correspond to the "large" architecture ($G = 20$, three hidden layers with 32 neurons each).

Table 3: Results on the Feynman benchmark for the "small" architecture ($G = 5$, two hidden layers with 8 neurons each). Reported values correspond to median final training loss and relative $L^2$ error with respect to the reference solution. Best results per equation are shown in bold. The power-law initialization uses $\alpha = 0.25, \beta = 1.75$.

| Function | Baseline | | Glorot | | Power-Law | |
|---|---|---|---|---|---|---|
| | Loss | $L^2$ | Loss | $L^2$ | Loss | $L^2$ |
| I.6.2 | $5.17 \cdot 10^{-3}$ | $\mathbf{4.05 \cdot 10^{-1}}$ | $9.86 \cdot 10^{-3}$ | $4.22 \cdot 10^{-1}$ | $\mathbf{1.18 \cdot 10^{-3}}$ | $4.14 \cdot 10^{-1}$ |
| I.6.2b | $3.58 \cdot 10^{-3}$ | $\mathbf{4.28 \cdot 10^{-1}}$ | $8.69 \cdot 10^{-3}$ | $5.01 \cdot 10^{-1}$ | $\mathbf{1.97 \cdot 10^{-3}}$ | $4.37 \cdot 10^{-1}$ |
| I.12.11 | $1.40 \cdot 10^{-5}$ | $3.67 \cdot 10^{-3}$ | $1.30 \cdot 10^{-5}$ | $3.75 \cdot 10^{-3}$ | $\mathbf{1.12 \cdot 10^{-6}}$ | $\mathbf{1.07 \cdot 10^{-3}}$ |
| I.13.12 | $2.24 \cdot 10^{3}$ | $1.86 \cdot 10^{0}$ | $3.51 \cdot 10^{3}$ | $\mathbf{7.65 \cdot 10^{-1}}$ | $1.75 \cdot 10^{3}$ | $2.36 \cdot 10^{0}$ |
| I.16.6 | $2.62 \cdot 10^{-4}$ | $3.55 \cdot 10^{-2}$ | $2.94 \cdot 10^{-4}$ | $3.63 \cdot 10^{-2}$ | $\mathbf{1.19 \cdot 10^{-4}}$ | $\mathbf{2.92 \cdot 10^{-2}}$ |
| I.18.4 | $1.39 \cdot 10^{3}$ | $\mathbf{1.00 \cdot 10^{0}}$ | $2.31 \cdot 10^{3}$ | $\mathbf{1.00 \cdot 10^{0}}$ | $1.04 \cdot 10^{3}$ | $\mathbf{1.00 \cdot 10^{0}}$ |
| I.26.2 | $5.00 \cdot 10^{-6}$ | $7.21 \cdot 10^{-3}$ | $1.40 \cdot 10^{-5}$ | $1.19 \cdot 10^{-2}$ | $\mathbf{9.99 \cdot 10^{-7}}$ | $\mathbf{3.13 \cdot 10^{-3}}$ |
| I.27.6 | $\mathbf{1.87 \cdot 10^{-3}}$ | $\mathbf{1.00 \cdot 10^{0}}$ | $1.24 \cdot 10^{-1}$ | $\mathbf{1.00 \cdot 10^{0}}$ | $1.77 \cdot 10^{-1}$ | $\mathbf{1.00 \cdot 10^{0}}$ |
| I.29.16 | $1.05 \cdot 10^{-4}$ | $1.14 \cdot 10^{-2}$ | $1.22 \cdot 10^{-4}$ | $1.24 \cdot 10^{-2}$ | $\mathbf{3.14 \cdot 10^{-5}}$ | $\mathbf{6.83 \cdot 10^{-3}}$ |
| I.30.3 | $4.00 \cdot 10^{-6}$ | $4.62 \cdot 10^{-3}$ | $9.00 \cdot 10^{-6}$ | $6.84 \cdot 10^{-3}$ | $\mathbf{4.88 \cdot 10^{-7}}$ | $\mathbf{1.73 \cdot 10^{-3}}$ |
| I.40.1 | $1.30 \cdot 10^{-5}$ | $4.76 \cdot 10^{-3}$ | $3.90 \cdot 10^{-5}$ | $8.11 \cdot 10^{-3}$ | $\mathbf{1.74 \cdot 10^{-6}}$ | $\mathbf{1.81 \cdot 10^{-3}}$ |
| I.50.26 | $1.40 \cdot 10^{-5}$ | $4.07 \cdot 10^{-3}$ | $1.00 \cdot 10^{-5}$ | $3.47 \cdot 10^{-3}$ | $\mathbf{1.17 \cdot 10^{-6}}$ | $\mathbf{1.20 \cdot 10^{-3}}$ |
| II.2.42 | $1.52 \cdot 10^{-4}$ | $4.46 \cdot 10^{-3}$ | $2.50 \cdot 10^{-5}$ | $7.74 \cdot 10^{-3}$ | $\mathbf{8.49 \cdot 10^{-7}}$ | $\mathbf{1.44 \cdot 10^{-3}}$ |
| II.6.15a | $6.00 \cdot 10^{-6}$ | $7.35 \cdot 10^{-2}$ | $1.80 \cdot 10^{-5}$ | $1.16 \cdot 10^{-1}$ | $\mathbf{4.97 \cdot 10^{-7}}$ | $\mathbf{1.89 \cdot 10^{-2}}$ |
| II.11.7 | $2.70 \cdot 10^{-5}$ | $1.03 \cdot 10^{-2}$ | $6.10 \cdot 10^{-5}$ | $1.44 \cdot 10^{-2}$ | $\mathbf{3.58 \cdot 10^{-6}}$ | $\mathbf{4.08 \cdot 10^{-3}}$ |
| II.11.27 | $4.00 \cdot 10^{-6}$ | $6.20 \cdot 10^{-3}$ | $1.50 \cdot 10^{-5}$ | $1.21 \cdot 10^{-2}$ | $\mathbf{7.17 \cdot 10^{-7}}$ | $\mathbf{2.72 \cdot 10^{-3}}$ |
| II.35.18 | $3.00 \cdot 10^{-6}$ | $7.61 \cdot 10^{-3}$ | $1.10 \cdot 10^{-5}$ | $1.38 \cdot 10^{-2}$ | $\mathbf{1.84 \cdot 10^{-7}}$ | $\mathbf{1.48 \cdot 10^{-3}}$ |
| II.36.38 | $3.50 \cdot 10^{-5}$ | $1.17 \cdot 10^{-2}$ | $6.50 \cdot 10^{-5}$ | $1.57 \cdot 10^{-2}$ | $\mathbf{2.71 \cdot 10^{-6}}$ | $\mathbf{3.43 \cdot 10^{-3}}$ |
| III.10.19 | $1.40 \cdot 10^{-5}$ | $3.14 \cdot 10^{-3}$ | $1.50 \cdot 10^{-5}$ | $2.90 \cdot 10^{-3}$ | $\mathbf{8.26 \cdot 10^{-6}}$ | $\mathbf{2.24 \cdot 10^{-3}}$ |
| III.17.37 | $2.60 \cdot 10^{-5}$ | $1.05 \cdot 10^{-2}$ | $5.30 \cdot 10^{-5}$ | $1.41 \cdot 10^{-2}$ | $\mathbf{4.35 \cdot 10^{-6}}$ | $\mathbf{4.37 \cdot 10^{-3}}$ |

The results confirm the same overall trends observed in the earlier benchmarks. In both settings, power-law initialization achieves the best performance on the majority of equations, often by large margins in terms of both final training loss and relative $L^2$ error. Glorot initialization also provides substantial improvements over the baseline, particularly in the large architecture, where it consistently narrows the gap to power-law. A comparison between Tables 3 and 4 further highlights the role of initialization: with Glorot and power-law, the richer architecture is able to drive the loss down by several orders of magnitude and simultaneously reduce the $L^2$ error, whereas under the baseline initialization, performance often degrades when moving from the small to the large setting.

Table 4: Results on the Feynman benchmark for the "large" architecture ($G = 20$, three hidden layers with 32 neurons each). Reported values correspond to median final training loss and relative $L^2$ error with respect to the reference solution. Best results per equation are shown in bold. The power-law initialization uses $\alpha = 0.25, \beta = 1.75$.

| Function | Baseline Loss | Baseline $L^2$ | Glorot Loss | Glorot $L^2$ | Power-Law Loss | Power-Law $L^2$ |
|---|---|---|---|---|---|---|
| I.6.2 | $1.09 \cdot 10^{-3}$ | $1.51 \cdot 10^{0}$ | $4.80 \cdot 10^{-5}$ | $4.19 \cdot 10^{-1}$ | $\mathbf{5.20 \cdot 10^{-6}}$ | $\mathbf{3.85 \cdot 10^{-1}}$ |
| I.6.2b | $1.36 \cdot 10^{-3}$ | $1.64 \cdot 10^{0}$ | $7.60 \cdot 10^{-5}$ | $5.80 \cdot 10^{-1}$ | $\mathbf{2.18 \cdot 10^{-6}}$ | $\mathbf{4.59 \cdot 10^{-1}}$ |
| I.12.11 | $1.64 \cdot 10^{-4}$ | $3.77 \cdot 10^{-1}$ | $3.00 \cdot 10^{-6}$ | $1.47 \cdot 10^{-3}$ | $\mathbf{2.16 \cdot 10^{-8}}$ | $\mathbf{1.66 \cdot 10^{-4}}$ |
| I.13.12 | $2.70 \cdot 10^{3}$ | $3.08 \cdot 10^{0}$ | $2.81 \cdot 10^{3}$ | $\mathbf{1.11 \cdot 10^{0}}$ | $2.53 \cdot 10^{-1}$ | $5.49 \cdot 10^{0}$ |
| I.16.6 | $1.63 \cdot 10^{-4}$ | $6.31 \cdot 10^{-1}$ | $6.00 \cdot 10^{-6}$ | $1.63 \cdot 10^{-2}$ | $\mathbf{1.09 \cdot 10^{-6}}$ | $\mathbf{1.48 \cdot 10^{-2}}$ |
| I.18.4 | $2.67 \cdot 10^{2}$ | $\mathbf{1.00 \cdot 10^{0}}$ | $1.53 \cdot 10^{3}$ | $\mathbf{1.00 \cdot 10^{0}}$ | $\mathbf{4.15 \cdot 10^{-2}}$ | $\mathbf{1.00 \cdot 10^{0}}$ |
| I.26.2 | $1.01 \cdot 10^{-4}$ | $1.10 \cdot 10^{0}$ | $7.00 \cdot 10^{-6}$ | $8.98 \cdot 10^{-3}$ | $\mathbf{1.72 \cdot 10^{-7}}$ | $\mathbf{1.25 \cdot 10^{-3}}$ |
| I.27.6 | $3.33 \cdot 10^{-3}$ | $\mathbf{1.00 \cdot 10^{0}}$ | $1.85 \cdot 10^{-4}$ | $\mathbf{1.00 \cdot 10^{0}}$ | $\mathbf{8.93 \cdot 10^{-5}}$ | $\mathbf{1.00 \cdot 10^{0}}$ |
| I.29.16 | $2.01 \cdot 10^{-4}$ | $4.45 \cdot 10^{-1}$ | $1.20 \cdot 10^{-5}$ | $6.28 \cdot 10^{-3}$ | $\mathbf{2.06 \cdot 10^{-7}}$ | $\mathbf{2.57 \cdot 10^{-3}}$ |
| I.30.3 | $1.18 \cdot 10^{-4}$ | $7.72 \cdot 10^{-1}$ | $1.00 \cdot 10^{-6}$ | $2.92 \cdot 10^{-3}$ | $\mathbf{2.17 \cdot 10^{-8}}$ | $\mathbf{4.17 \cdot 10^{-4}}$ |
| I.40.1 | $2.26 \cdot 10^{-4}$ | $6.70 \cdot 10^{-1}$ | $5.00 \cdot 10^{-6}$ | $3.39 \cdot 10^{-3}$ | $\mathbf{1.41 \cdot 10^{-7}}$ | $\mathbf{6.17 \cdot 10^{-4}}$ |
| I.50.26 | $2.03 \cdot 10^{-4}$ | $4.38 \cdot 10^{-1}$ | $2.00 \cdot 10^{-6}$ | $1.50 \cdot 10^{-3}$ | $\mathbf{3.70 \cdot 10^{-8}}$ | $\mathbf{2.25 \cdot 10^{-4}}$ |
| II.2.42 | $1.52 \cdot 10^{-4}$ | $6.86 \cdot 10^{-1}$ | $4.00 \cdot 10^{-6}$ | $2.62 \cdot 10^{-3}$ | $\mathbf{8.54 \cdot 10^{-8}}$ | $\mathbf{4.98 \cdot 10^{-4}}$ |
| II.6.15a | $6.60 \cdot 10^{-5}$ | $7.60 \cdot 10^{0}$ | $2.00 \cdot 10^{-6}$ | $5.47 \cdot 10^{-2}$ | $\mathbf{8.13 \cdot 10^{-9}}$ | $\mathbf{4.40 \cdot 10^{-3}}$ |
| II.11.7 | $1.75 \cdot 10^{-4}$ | $9.78 \cdot 10^{-1}$ | $1.10 \cdot 10^{-5}$ | $1.01 \cdot 10^{-2}$ | $\mathbf{1.80 \cdot 10^{-7}}$ | $\mathbf{3.00 \cdot 10^{-3}}$ |
| II.11.27 | $8.80 \cdot 10^{-5}$ | $1.04 \cdot 10^{0}$ | $1.00 \cdot 10^{-6}$ | $3.76 \cdot 10^{-3}$ | $\mathbf{1.54 \cdot 10^{-7}}$ | $\mathbf{1.95 \cdot 10^{-3}}$ |
| II.35.18 | $7.40 \cdot 10^{-5}$ | $1.19 \cdot 10^{0}$ | $6.00 \cdot 10^{-6}$ | $1.18 \cdot 10^{-2}$ | $\mathbf{2.95 \cdot 10^{-8}}$ | $\mathbf{7.77 \cdot 10^{-4}}$ |
| II.36.38 | $1.93 \cdot 10^{-4}$ | $9.48 \cdot 10^{-1}$ | $8.00 \cdot 10^{-6}$ | $1.11 \cdot 10^{-2}$ | $\mathbf{3.05 \cdot 10^{-7}}$ | $\mathbf{4.92 \cdot 10^{-3}}$ |
| III.10.19 | $1.81 \cdot 10^{-4}$ | $2.74 \cdot 10^{-1}$ | $1.00 \cdot 10^{-6}$ | $9.89 \cdot 10^{-4}$ | $\mathbf{9.87 \cdot 10^{-9}}$ | $\mathbf{8.70 \cdot 10^{-5}}$ |
| III.17.37 | $1.45 \cdot 10^{-4}$ | $9.10 \cdot 10^{-1}$ | $4.90 \cdot 10^{-5}$ | $1.31 \cdot 10^{-2}$ | $\mathbf{6.45 \cdot 10^{-6}}$ | $\mathbf{5.14 \cdot 10^{-3}}$ |

## 5 CONCLUSION

In this work, we proposed and systematically evaluated new initialization strategies for spline-based KANs. Specifically, we introduced two theory-driven schemes inspired by LeCun and Glorot, including a variant with normalized basis functions, as well as an empirical family of power-law initializations. Through large-scale grid searches, we identified favorable exponent ranges for the power-law method for both function fitting and forward PDE problems. Across all evaluations, including loss curve analysis and NTK dynamics, we showed that initialization plays a critical role in KAN performance. In particular, our results demonstrate that while LeCun-inspired schemes offer limited benefits, Glorot-inspired initialization emerges as a strong candidate for parameter-rich architectures, and the empirical power-law family provides the most robust improvements overall, achieving faster convergence and lower errors across benchmarks. These findings highlight initialization as a key component of training KANs and identify effective practical strategies for the process.

### 5.1 LIMITATIONS AND FUTURE WORK

While our study establishes the importance of initialization in spline-based KANs, it also comes with limitations. Our power-law scheme, although empirically effective, currently lacks a rigorous theoretical foundation, and understanding why specific exponent ranges perform well remains an open question. Moreover, although we considered both supervised function fitting and physics-informed PDE benchmarks, further exploration in other domains such as reinforcement learning or generative modeling could provide additional insights. Addressing these limitations offers natural directions for future work, including deriving principled theory for power-law initialization, investigating transferability across KAN variants (e.g., Chebyshev-based or residual-free forms), and exploring initialization strategies in conjunction with adaptive optimization techniques.

## REPRODUCIBILITY STATEMENT

The full code (including selected seeds for each experiment) and the processed data from the grid-search experiments are publicly available at the following GitHub repository: `https://github.com/srigas/KAN_Initialization_Schemes`.

## ACKNOWLEDGMENTS

S. R. and G. A. are supported by the Innovative Health Initiative Joint Undertaking (IHI JU) under grant agreement No 101253520. The JU receives support from the European Union's Horizon Europe research and innovation programme and COCIR, EFPIA, Europa Bío, MedTech Europe, and Vaccines Europe.

## LLM USAGE

Large Language Models (LLMs) were used during peer review for grammar and syntax refinement only; all ideas, technical content, analyses and conclusions remain the authors' work.

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

## A  DERIVATION OF LECUN-INSPIRED INITIALIZATION SCHEME

In this appendix, we provide a derivation of Eqs. (2) from the main text. Assuming statistical independence between each term in the outer sum of Eq. (1) and requiring the output variance to match the input variance, one finds

$$\text{Var}(x_i) = n_{\text{in}}\text{Var}\left[r_{ji}\, R(x_i) + c_{ji} \sum_{m=1}^{G+k} b_{jim}\, B_m(x_i)\right], \tag{8}$$

where the right-hand side contains the variance of a sum of $G + k + 1$ terms: one residual term and $G+k$ spline basis terms. We adopt a simplifying assumption that the total variance is approximately equipartitioned across all components,[3] allowing us to bypass pairwise covariance terms. This leads to the following expressions for the residual and spline basis terms, respectively:

$$\frac{\text{Var}(x_i)}{G+k+1} = n_{\text{in}}\text{Var}\left[r_{ji}\, R(x_i)\right], \qquad \frac{\text{Var}(x_i)}{G+k+1} = n_{\text{in}}\text{Var}\left[b_{jim}\, B_m(x_i)\right]. \tag{9}$$

Since the trainable weights $r_{ji}$ are independent of the residual function $R(x_i)$, the variance of their product becomes

$$\text{Var}\left[r_{ji}\, R(x_i)\right] = \underbrace{\mathbb{E}^2(r_{ji})}_{=0}\text{Var}\left[R(x_i)\right] + \mathbb{E}^2\left[R(x_i)\right]\text{Var}(r_{ji}) + \text{Var}(r_{ji})\,\text{Var}\left[R(x_i)\right]$$

$$= \text{Var}(r_{ji})\left\{\text{Var}\left[R(x_i)\right] + \mathbb{E}^2\left[R(x_i)\right]\right\} = \sigma_r^2\,\mathbb{E}\left[R^2(x_i)\right] \tag{10}$$

and, in a completely analogous manner, we find

$$\text{Var}\left[b_{jim}\, B_m(x_i)\right] = \sigma_b^2\,\mathbb{E}\left[B_m^2(x_i)\right]. \tag{11}$$

Substitution of the expressions of Eqs. (10), (11) into Eqs. (9) yields Eq. (2) from Section 3.1.

---

[3]This assumption does not necessarily hold in general. For example, one could consider a 50%–50% split between the residual and basis function terms. We experimented with this alternative and found that it yielded poorer results compared to the variance partitioning that leads to Eqs. (2).

## B   DERIVATION OF GLOROT-INSPIRED INITIALIZATION SCHEME

In this appendix, we derive Eqs. (5) from the main text. Unlike the LeCun-inspired scheme, which focuses solely on variance preservation in the forward pass, the Glorot principle (Glorot & Bengio, 2010) requires that the variance of both activations and backpropagated gradients remain constant across layers. For analytical tractability, and following the standard assumption in this setting, we further approximate these constant values by unity,

$$\text{Var}(x_i) = \text{Var}(y_i) \approx 1, \qquad \text{Var}(\delta x_i) = \text{Var}(\delta y_i) \approx 1, \tag{12}$$

where $\delta x_i = \partial \mathcal{L}/\partial x_i$ and $\delta y_j = \partial \mathcal{L}/\partial y_j$, with $\mathcal{L}$ denoting the loss function. This approximation is consistent with the common assumption of i.i.d. inputs with zero mean and unit variance. In practice, when this assumption does not hold, an additional gain factor can be introduced to rescale the initialization, as is standard in frameworks such as `PyTorch`.

Using the result from Appendix A together with the first condition of Eq. (12), the constraints for variance preservation in the forward pass can be written as

$$1 = (G + k + 1)\, n_{\text{in}}\, \sigma_r^2\, \mu_R^{(0)}, \qquad 1 = (G + k + 1)\, n_{\text{in}}\, \sigma_b^2\, \mu_B^{(0)}, \tag{13}$$

where $\mu_R^{(0)} = \mathbb{E}[R(x_i)^2]$ and $\mu_B^{(0)} = \mathbb{E}[B_m(x_i)^2]$ as defined in Eq. (3).

For the backward pass, differentiating Eq. (1) with respect to $x_i$ gives

$$\frac{\partial y_j}{\partial x_i} \;=\; r_{ji}\, R'(x_i) \;+\; c_{ji} \sum_{m=1}^{G+k} b_{jim}\, B'_m(x_i), \tag{14}$$

Setting $c_{ji} = 1$, the chain rule yields

$$\delta x_i \;=\; \sum_{j=1}^{n_{\text{out}}} \frac{\partial y_j}{\partial x_i}\, \delta y_j \;=\; \underbrace{\sum_{j=1}^{n_{\text{out}}} r_{ji}\, R'(x_i)\, \delta y_j}_{\text{residual contribution}} \;+\; \sum_{m=1}^{G+k} \underbrace{\sum_{j=1}^{n_{\text{out}}} b_{jim}\, B'_m(x_i)\, \delta y_j}_{m\text{-th spline contribution}}, \tag{15}$$

and applying the second condition of Eq. (12) gives

$$1 \;=\; n_{\text{out}} \text{Var}\left[ r_{ji}\, R'(x_i) + \sum_{m=1}^{G+k} b_{jim}\, B'_m(x_i) \right], \tag{16}$$

where we have adopted the standard Glorot assumptions: the $\delta y_j$ are zero-mean, mutually independent, and independent of weights and inputs. At this point we may again equipartition the total variance across the $(G+k+1)$ components (one residual term and $G+k$ spline terms), exactly mirroring the forward-pass treatment. This leads to

$$1 = (G + k + 1)\, n_{\text{out}}\, \text{Var}\left[ r_{ji}\, R'(x_i) \right], \qquad 1 = (G + k + 1)\, n_{\text{out}}\, \text{Var}\left[ b_{jim}\, B'_m(x_i) \right], \tag{17}$$

and, following the same arguments as in Appendix A, we find

$$1 = (G + k + 1)\, n_{\text{out}}\, \sigma_r^2\, \mu_R^{(1)}, \qquad 1 = (G + k + 1)\, n_{\text{out}}\, \sigma_b^2\, \mu_B^{(1)}, \tag{18}$$

where $\mu_R^{(1)} = \mathbb{E}[R'(x_i)^2]$ and $\mu_B^{(1)} = \mathbb{E}[B'_m(x_i)^2]$ as defined in Eq. (6).

Equations (13) and (17) are the forward- and backward-pass constraints, respectively. Balancing them in the Glorot manner (i.e., by harmonic averaging) yields the standard deviations in Eq. (5) of

the main text. As a sanity check, consider an MLP: the residual term is absent, and the linear layer followed by a nonlinearity can be viewed as a single basis function. For the common hyperbolic tangent activation, $\mu_B^{(0)} \approx \mu_B^{(1)} \approx 1$ (Glorot & Bengio, 2010), so our scheme reduces to

$$\sigma_b = \sqrt{\underbrace{\frac{1}{G+k+1}}_{=1} \cdot \frac{2}{n_{\text{in}} \underbrace{\mu_B^{(0)}}_{\approx 1} + n_{\text{out}} \underbrace{\mu_B^{(1)}}_{\approx 1}}} = \sqrt{\frac{2}{n_{\text{in}} + n_{\text{out}}}}, \tag{19}$$

which recovers the classical Glorot initialization.

## C   IMPLEMENTATION DETAILS

This appendix provides the full specifications of the benchmarks used in our experiments, including the functional forms of the target problems, training setups, and data generation procedures. We separate the discussion into three parts: function fitting, forward PDE problems and the Feynman dataset.

### C.1   FUNCTION FITTING

For the function fitting experiments of Section 4.1 and Section 4.2, we study five two-dimensional functions ranging from simple expressions to more complex, nonlinear, or piecewise-defined forms. Specifically, we consider the following functions in the $[-1, 1] \times [-1, 1]$ domain:

- $f_1(x, y) = xy$
- $f_2(x, y) = \exp\left(\sin(\pi x) + y^2\right)$
- $f_3(x, y) = I_1(x) + \exp\left[\exp\left(-|y|\right) I_1(y)\right] + \sin(xy)$
- $f_4(x, y) = S\left[f_3(x, y) + \mathrm{erf}^{-1}(y)\right] \times C\left[f_3(x, y) + \mathrm{erf}^{-1}(y)\right]$
- $f_5(x, y) = y \cdot \mathrm{sgn}(0.5 - x) + \mathrm{erf}(x) \cdot \min\left(xy, \frac{1}{xy}\right)$

where $I_1(x)$ is the modified Bessel function of first order, $\mathrm{sgn}(x)$ is the sign function, $\mathrm{erf}(x)$ is the error function and $S(x)$, $C(x)$ are the Fresnel integral functions defined as

$$S(x) = \int_0^x \sin\left(\frac{\pi t^2}{2}\right) dt, \qquad C(x) = \int_0^x \cos\left(\frac{\pi t^2}{2}\right) dt. \tag{20}$$

The reference surfaces for these functions are shown in Figure 5.

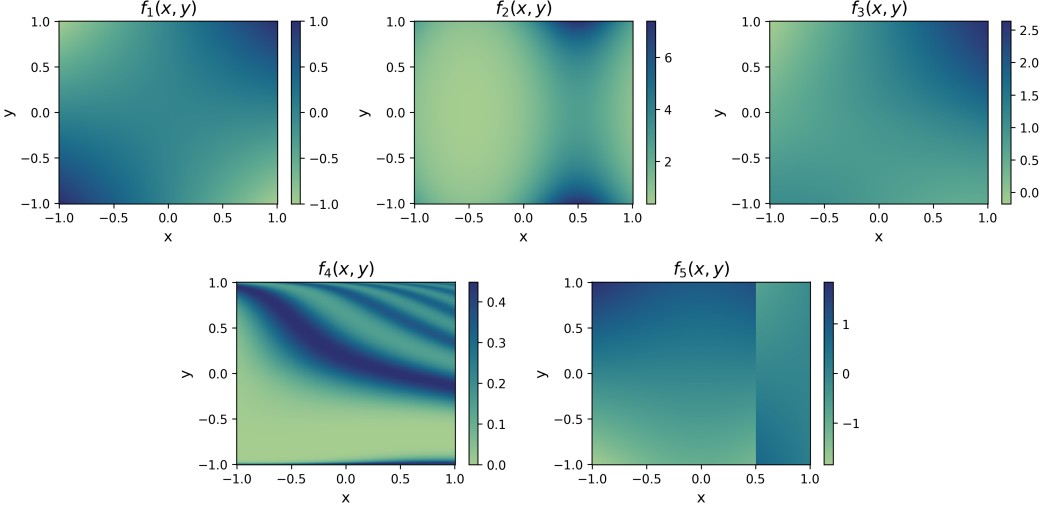

Figure 5: Reference surfaces for the five two-dimensional target functions $f_1$ through $f_5$ used in the function fitting experiments.

The KAN models used to fit these functions utilize spline basis functions of order $k = 3$, defined over an augmented, uniform grid within the $[-1, 1]$ domain (Liu et al., 2025). Training is performed using the Adam optimizer with a fixed learning rate of $10^{-3}$, with the objective of minimizing the mean squared error between the predicted and reference function values. For each target function $f_i(x, y)$, with $i = 1, \ldots, 5$, we generate 4,000 random input samples uniformly distributed over the

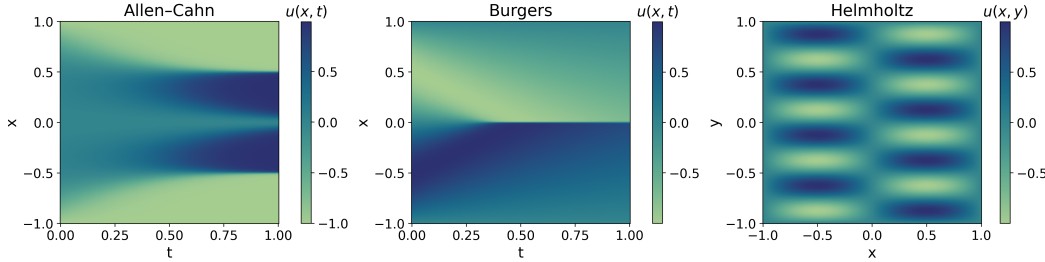

Figure 6: Reference solutions for the three PDE problems considered.

domain $[-1, 1] \times [-1, 1]$, and calculate the corresponding outputs to serve as ground truth during training. To compute the relative $L^2$ error between the model predictions and reference solutions, we evaluate all trained models on a uniform $200 \times 200$ grid covering the same domain.

## C.2 FORWARD PDE PROBLEMS

In addition to function fitting, we consider three representative forward PDEs commonly used as PIML benchmarks. For each case, we specify the governing equation, domain and boundary/initial conditions.

**Allen–Cahn equation.** We solve the Allen–Cahn equation on $(t, x) \in [0, 1] \times [-1, 1]$:

$$u_t(t, x) - D\, u_{xx}(t, x) - c\big(u(t, x) - u(t, x)^3\big) \;=\; 0, \tag{21}$$

with diffusion coefficient $D = 10^{-4}$ and reaction strength $c = 5$. The initial and boundary conditions are

$$u(0, x) \;=\; x^2 \cos(\pi x), \qquad x \in [-1, 1], \tag{22}$$

$$u(t, -1) \;=\; u(t, 1) \;=\; -1, \qquad t \in [0, 1]. \tag{23}$$

Since the Allen–Cahn equation has no analytic closed-form solution, we use the reference solution used in Wang et al. (2024), which is depicted in the left plot of Fig. 6.

**Burgers' equation.** We solve the viscous Burgers' equation on $(t, x) \in [0, 1] \times [-1, 1]$:

$$u_t(t, x) + u(t, x)\, u_x(t, x) - \nu\, u_{xx}(t, x) \;=\; 0, \tag{24}$$

for $\nu = 0.01/\pi$, with initial and boundary conditions

$$u(0, x) \;=\; -\sin(\pi x), \qquad x \in [-1, 1], \tag{25}$$

$$u(t, -1) \;=\; u(t, 1) \;=\; 0, \qquad t \in [0, 1]. \tag{26}$$

Similar to the Allen–Cahn equation, Burger's equation has no analytic closed-form solution, therefore we use the reference solution used in Wang et al. (2024), which is depicted in the middle plot of Fig. 6.

**Helmholtz equation.** We solve a two-dimensional Helmholtz problem on $(x, y) \in [-1, 1]^2$ with unit wavenumber and a separable sinusoidal source:

$$u_{xx}(x, y) + u_{yy}(x, y) + u(x, y) = f(x, y), \tag{27}$$

where

$$f(x, y) = \left(1 - \pi^2(a_1^2 + a_2^2)\right) \sin(\pi a_1 x) \sin(\pi a_2 y), \tag{28}$$

and $a_1 = 1$ and $a_2 = 4$. We consider homogeneous Dirichlet boundary conditions:

$$u(x, y) = 0 \quad \text{for} \quad (x, y) \in \partial([-1, 1]^2). \tag{29}$$

The analytic solution to this PDE problem is

$$u_{\text{ref}}(x, y) = \sin(\pi x) \sin(4\pi y), \tag{30}$$

and is depicted in the right plot of Fig. 6 for $x, y$ sampled on a uniform $512 \times 512$ grid.

The PDE problems are solved using the Residual-Based Attention (RBA) weighting scheme (Anagnostopoulos et al., 2024) within the PIML framework (Raissi et al., 2019), where the training objective is defined as a sum of weighted residuals associated with the PDE differential operator and the boundary/initial condition operators. Specifically, we minimize

$$\mathcal{L}(\boldsymbol{\theta}) = \frac{1}{N_{\text{pde}}} \sum_{i=1}^{N_{\text{pde}}} \left| \alpha_i^{(\text{pde})} r_i^{(\text{pde})}(\boldsymbol{\theta}) \right|^2 + \frac{1}{N_{\text{bc}}} \sum_{i=1}^{N_{\text{bc}}} \left| \alpha_i^{(\text{bc})} r_i^{(\text{bc})}(\boldsymbol{\theta}) \right|^2, \tag{31}$$

where $\|\cdot\|_2$ denotes the $L^2$ norm. Here, $r_i^{(\text{pde})}$ represents the residual of the governing PDE evaluated at the $i$-th collocation point, while $r_i^{(\text{bc})}$ denotes the residual of the boundary or initial condition (both are included in the second summation). The weights $\alpha_i^{(\xi)}$ ($\xi \in \{\text{pde}, \text{bc}\}$) are initialized to 1 and updated after each training iteration according to

$$\alpha_i^{(\xi), (\text{new})} = \gamma \, \alpha_i^{(\xi), (\text{old})} + \eta \frac{\left| r_i^{(\xi)} \right|}{\max_j \left\{ |r_j^{(\xi)}| \right\}_{j=1}^{N_\xi}}, \tag{32}$$

with hyperparameters $\gamma = 0.999$ and $\eta = 0.01$. This formulation ensures that collocation points with larger relative residuals are assigned greater importance during optimization[4].

We minimize the loss function in Eq. (31) using the Adam optimizer with a fixed learning rate of $10^{-3}$, operating in full-batch mode. For each PDE, we sample $N_{\text{pde}} = 2^{12}$ collocation points uniformly from a $2^6 \times 2^6$ grid, while for boundary and initial conditions we use $2^6$ collocation points per condition, sampled uniformly along the corresponding axis. The spline basis functions are defined as in the function fitting case (see Appendix C.1).

## C.3 FEYNMAN DATASET

As a third benchmark, we consider the subset of the Feynman dataset used in Section 4.3. The implementation details are identical to those of the function fitting benchmarks in Appendix C.1, with the exception of sampling. In this case, we generate 4,000 random input samples uniformly distributed over the domain $(-1, 0) \cup (0, 1)$, explicitly excluding the points $-1$, $0$, and $1$ to avoid singularities in certain formulas.

To compute the relative $L^2$ error between model predictions and reference solutions, we evaluate all trained models on a uniform $200 \times 200$ grid for two-dimensional functions and a uniform $30 \times 30 \times$

---

[4]Without RBA, the models trained to solve the Allen–Cahn equation would yield highly inaccurate solutions, preventing a meaningful comparison of initialization schemes.

30 grid for three-dimensional functions. Table 5 lists the indices of the functions included in this benchmark, together with their explicit expressions for reference.

Table 5: Dimensionless formulas from the Feynman dataset used in the benchmark. Each entry shows the dataset index and the corresponding explicit expression.

| Index | Formula |
|---|---|
| I.6.2 | $f_1(x_1, x_2) = \exp\left(-\frac{x_1^2}{2x_2^2}\right) \cdot \left(2\pi x_2^2\right)^{-1/2}$ |
| I.6.2b | $f_2(x_1, x_2, x_3) = \exp\left(-\frac{(x_1-x_2)^2}{2x_3^2}\right) \cdot \left(2\pi x_3^2\right)^{-1/2}$ |
| I.12.11 | $f_3(x_1, x_2) = 1 + x_1 \sin(x_2)$ |
| I.13.12 | $f_4(x_1, x_2) = x_1(1/x_2 - 1)$ |
| I.16.6 | $f_5(x_1, x_2) = (x_1 + x_2)/(1 + x_1 x_2)$ |
| I.18.4 | $f_6(x_1, x_2) = (1 + x_1 x_2)/(1 + x_1)$ |
| I.26.2 | $f_7(x_1, x_2) = \arcsin(x_1 \sin(x_2))$ |
| I.27.6 | $f_8(x_1, x_2) = 1/(1 + x_1 x_2)$ |
| I.29.16 | $f_9(x_1, x_2, x_3) = \sqrt{1 + x_1^2 - 2x_1 \cos(x_2 - x_3)}$ |
| I.30.3 | $f_{10}(x_1, x_2) = \sin^2(x_1 x_2/2)/\sin^2(x_2/2)$ |
| I.40.1 | $f_{11}(x_1, x_2) = x_1 \exp(-x_2)$ |
| I.50.26 | $f_{12}(x_1, x_2) = \cos(x_1) + x_2 \cos^2(x_1)$ |
| II.2.42 | $f_{13}(x_1, x_2) = (x_1 - 1)x_2$ |
| II.6.15a | $f_{14}(x_1, x_2, x_3) = \frac{x_3}{4\pi}\sqrt{x_1^2 + x_2^2}$ |
| II.11.7 | $f_{15}(x_1, x_2, x_3) = x_1(1 + x_2 \cos(x_3))$ |
| II.11.27 | $f_{16}(x_1, x_2) = (x_1 x_2)/(1 - \frac{x_1 x_2}{3})$ |
| II.35.18 | $f_{17}(x_1, x_2) = x_1/(\exp(x_2) + \exp(-x_2))$ |
| II.36.38 | $f_{18}(x_1, x_2, x_3) = x_1 + x_2 x_3$ |
| III.10.19 | $f_{19}(x_1, x_2) = \sqrt{1 + x_1^2 + x_2^2}$ |
| III.17.37 | $f_{20}(x_1, x_2, x_3) = x_2 (1 + x_1 \cos(x_3))$ |

## D    INDICATIVE RESULTS FOR POWER-LAW GRID-SEARCH

To illustrate the performance landscape of the power-law initialization, we present heatmaps over $(\alpha, \beta)$ configurations for representative cases. Specifically, Figures 7–10 show results for the function $f_3(x, y)$ across the four grid sizes, while Figures 11–13 show the corresponding results for the Allen–Cahn PDE. In each heatmap, the horizontal axis corresponds to $\alpha$ and the vertical axis to $\beta$, with rows and columns indicating different network widths and depths, respectively. These visualizations highlight the regions where power-law initialization provides the greatest improvements, and help motivate the choice of $(\alpha, \beta) = (0.25, 1.75)$ used for the architectures studied in Sections 4.2 and 4.3. Complete heatmaps for all benchmarks are included in the supplementary material (see Reproducibility Statement).

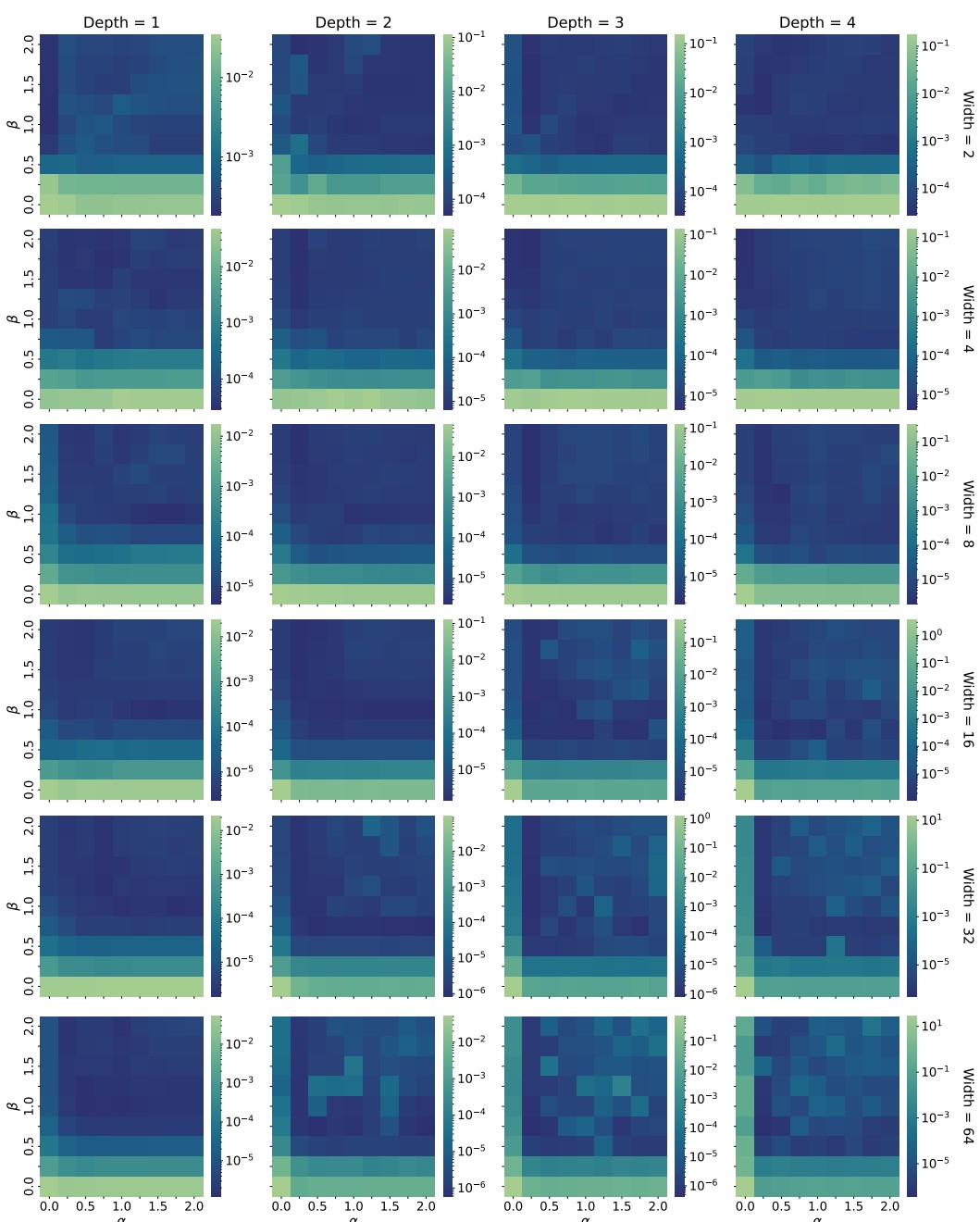

Figure 7: Grid search for the power-law initialization applied to fit function $f_3(x, y)$ for $G = 5$. Each heatmap corresponds to an architecture, with the horizontal and vertical axis representing $\alpha$ and $\beta$, respectively, and color denoting final training loss.

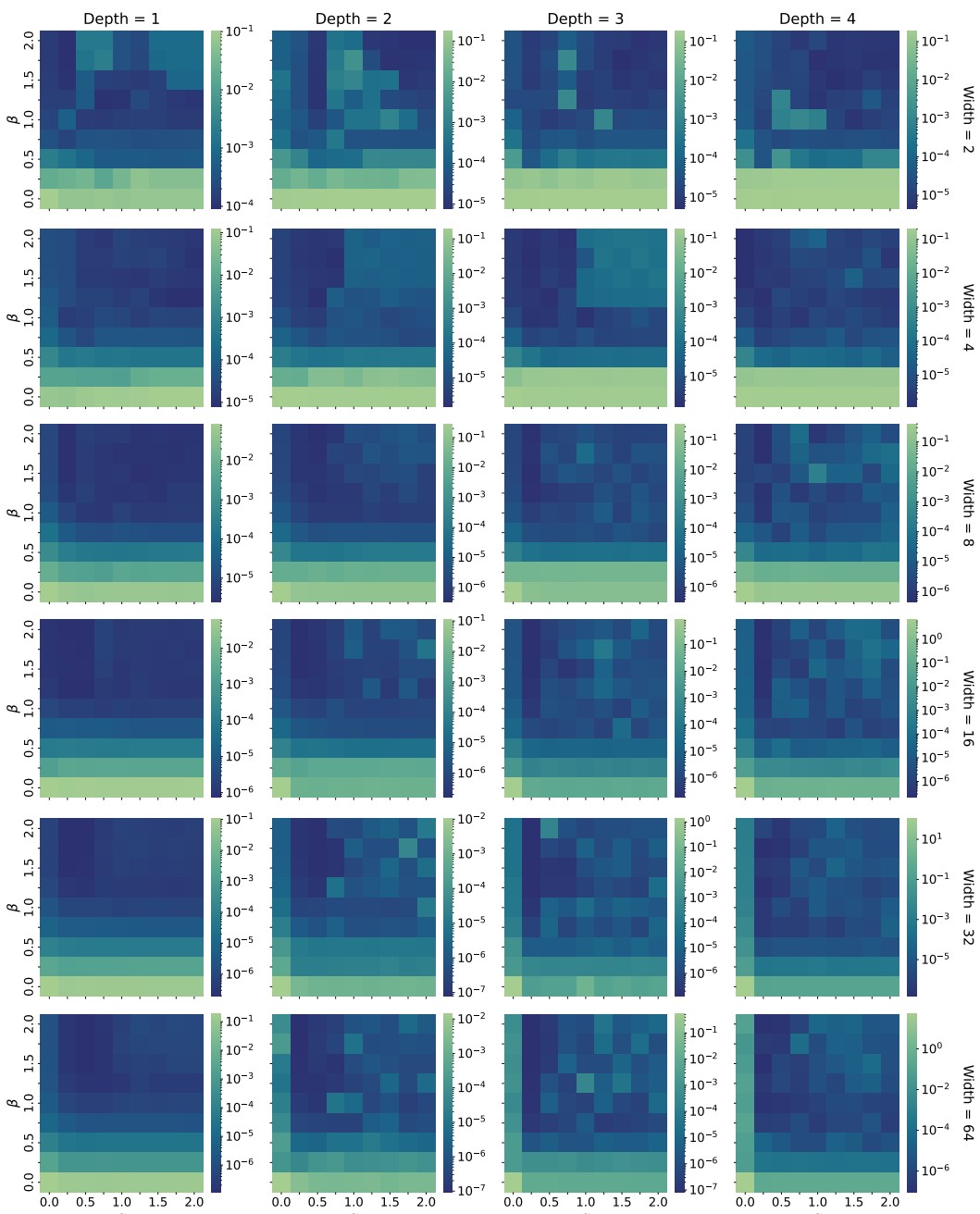

Figure 8: Grid search for the power-law initialization applied to fit function $f_3(x, y)$ for $G = 10$. Each heatmap corresponds to an architecture, with the horizontal and vertical axis representing $\alpha$ and $\beta$, respectively, and color denoting final training loss.

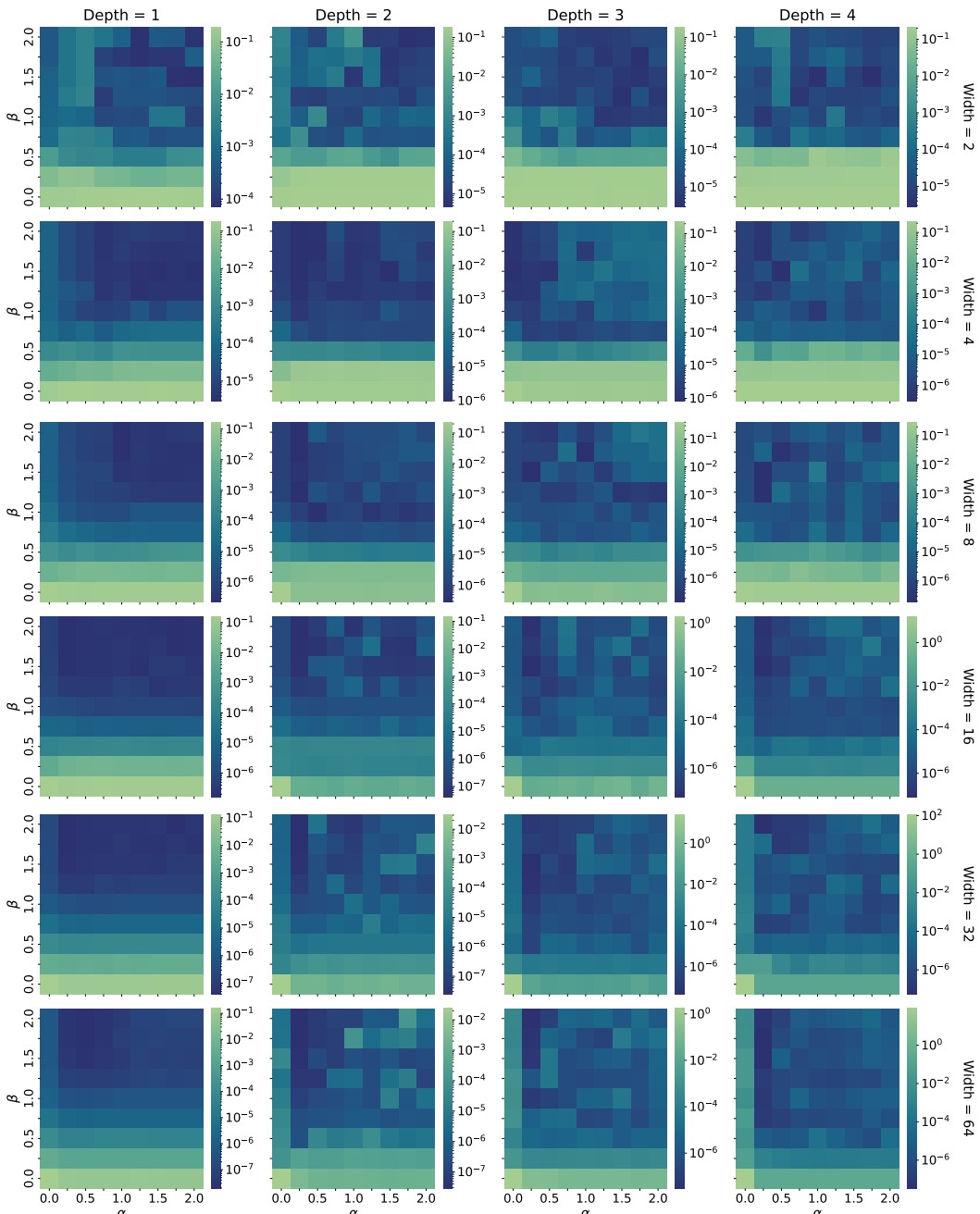

Figure 9: Grid search for the power-law initialization applied to fit function $f_3(x, y)$ for $G = 20$. Each heatmap corresponds to an architecture, with the horizontal and vertical axis representing $\alpha$ and $\beta$, respectively, and color denoting final training loss.

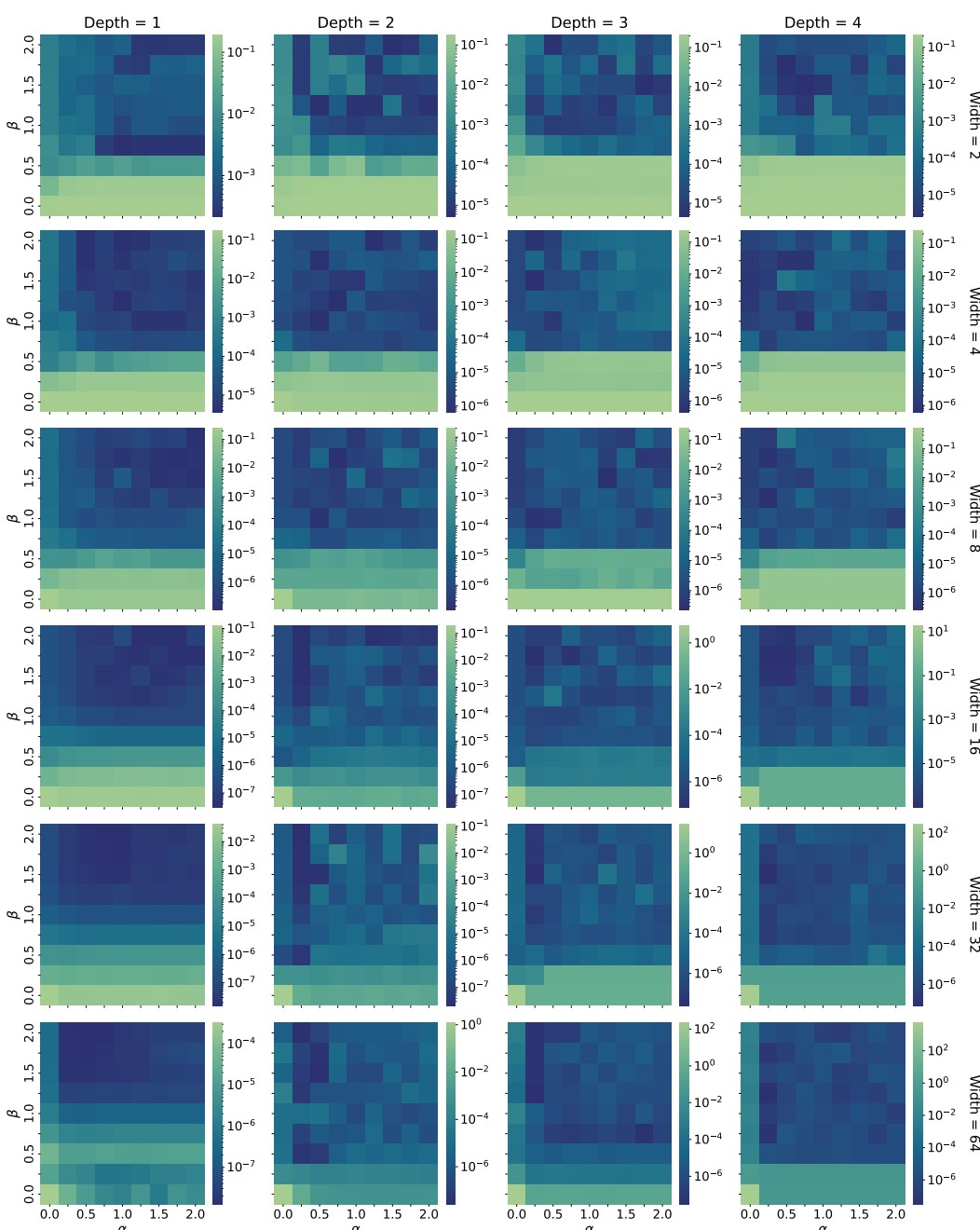

Figure 10: Grid search for the power-law initialization applied to fit function $f_3(x, y)$ for $G = 40$. Each heatmap corresponds to an architecture, with the horizontal and vertical axis representing $\alpha$ and $\beta$, respectively, and color denoting final training loss.

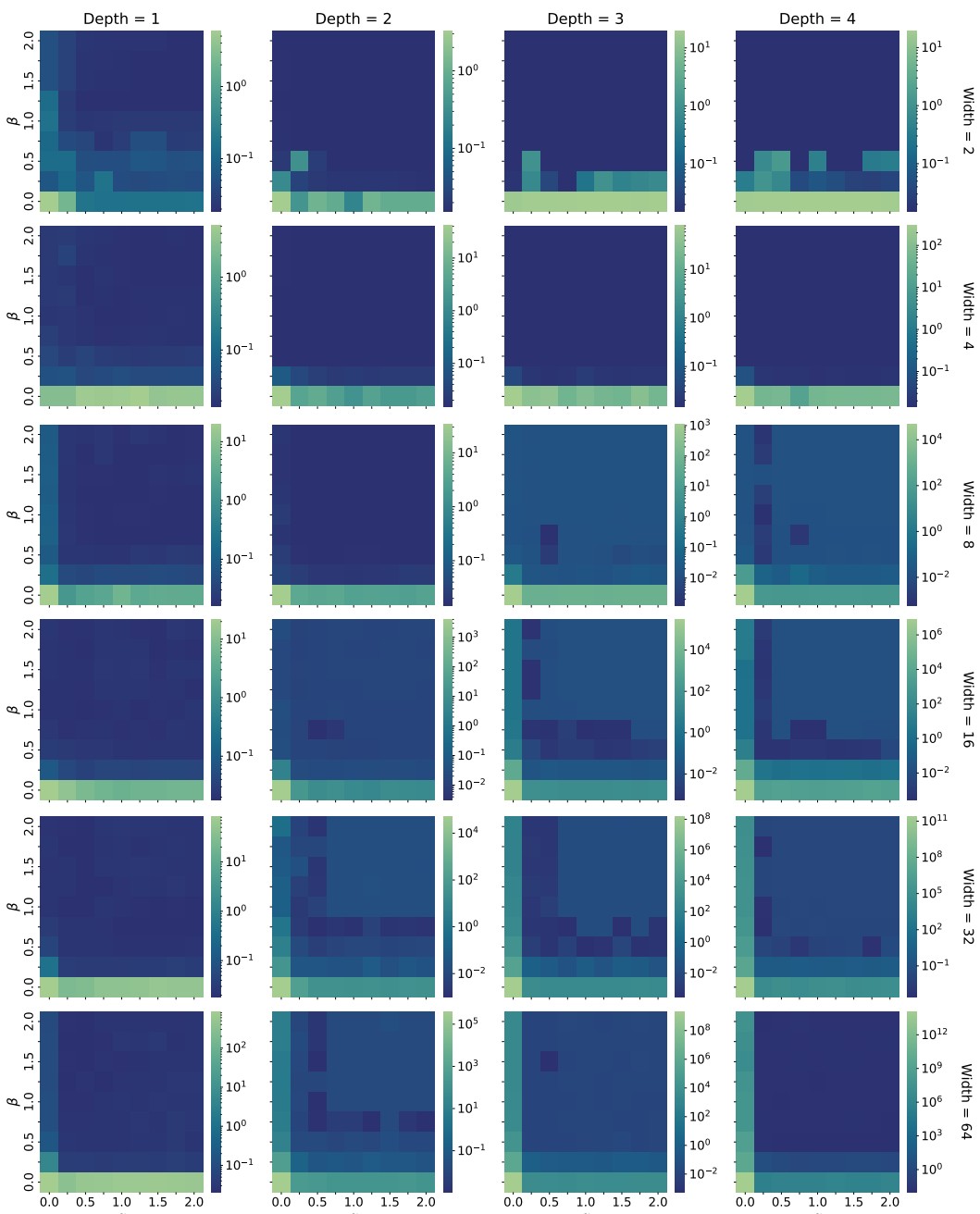

Figure 11: Grid search for the power-law initialization applied for the solution of the Allen–Cahn equation for $G = 5$. Each heatmap corresponds to an architecture, with the horizontal and vertical axis representing $\alpha$ and $\beta$, respectively, and color denoting final training loss.

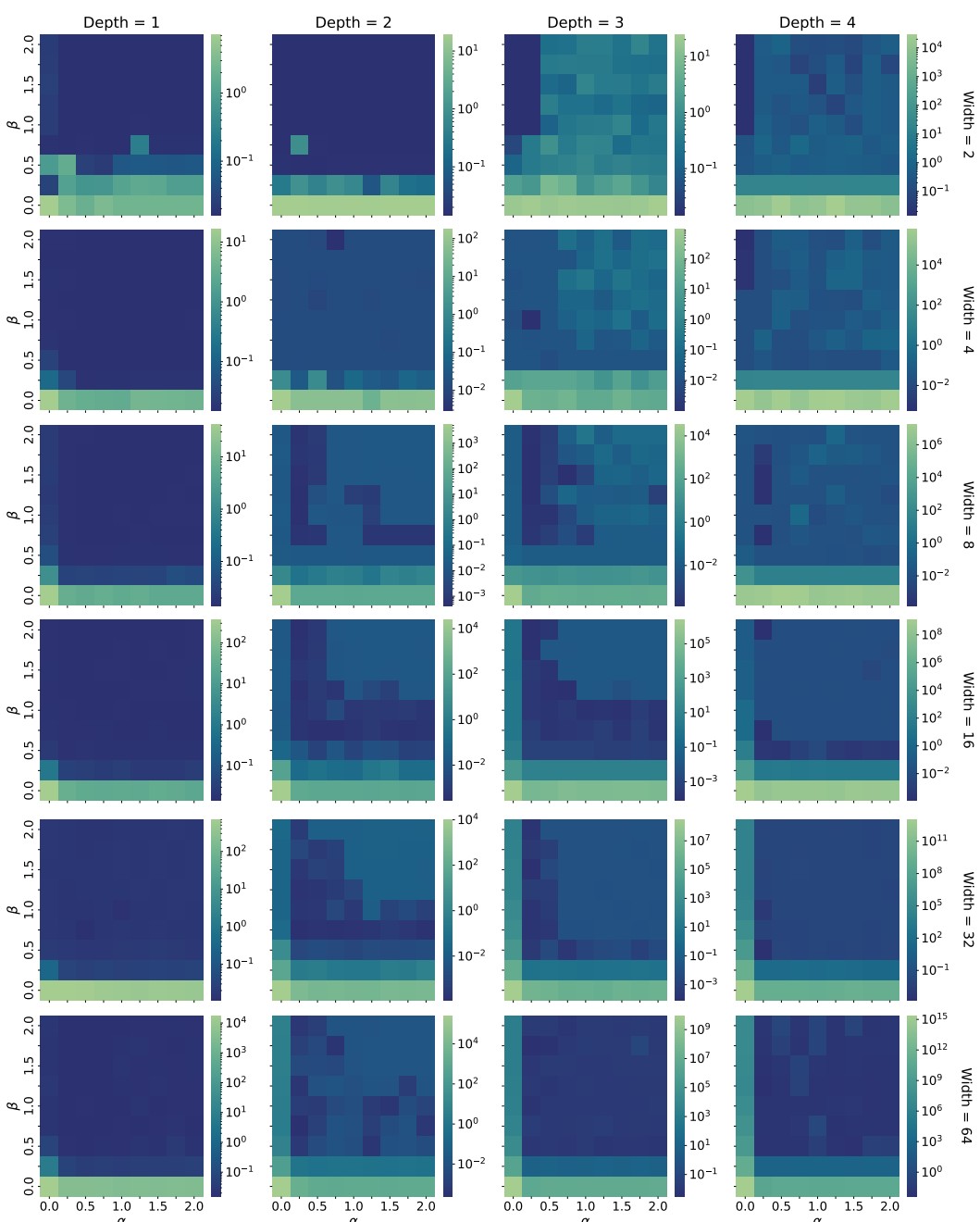

Figure 12: Grid search for the power-law initialization applied for the solution of the Allen–Cahn equation for $G = 10$. Each heatmap corresponds to an architecture, with the horizontal and vertical axis representing $\alpha$ and $\beta$, respectively, and color denoting final training loss.

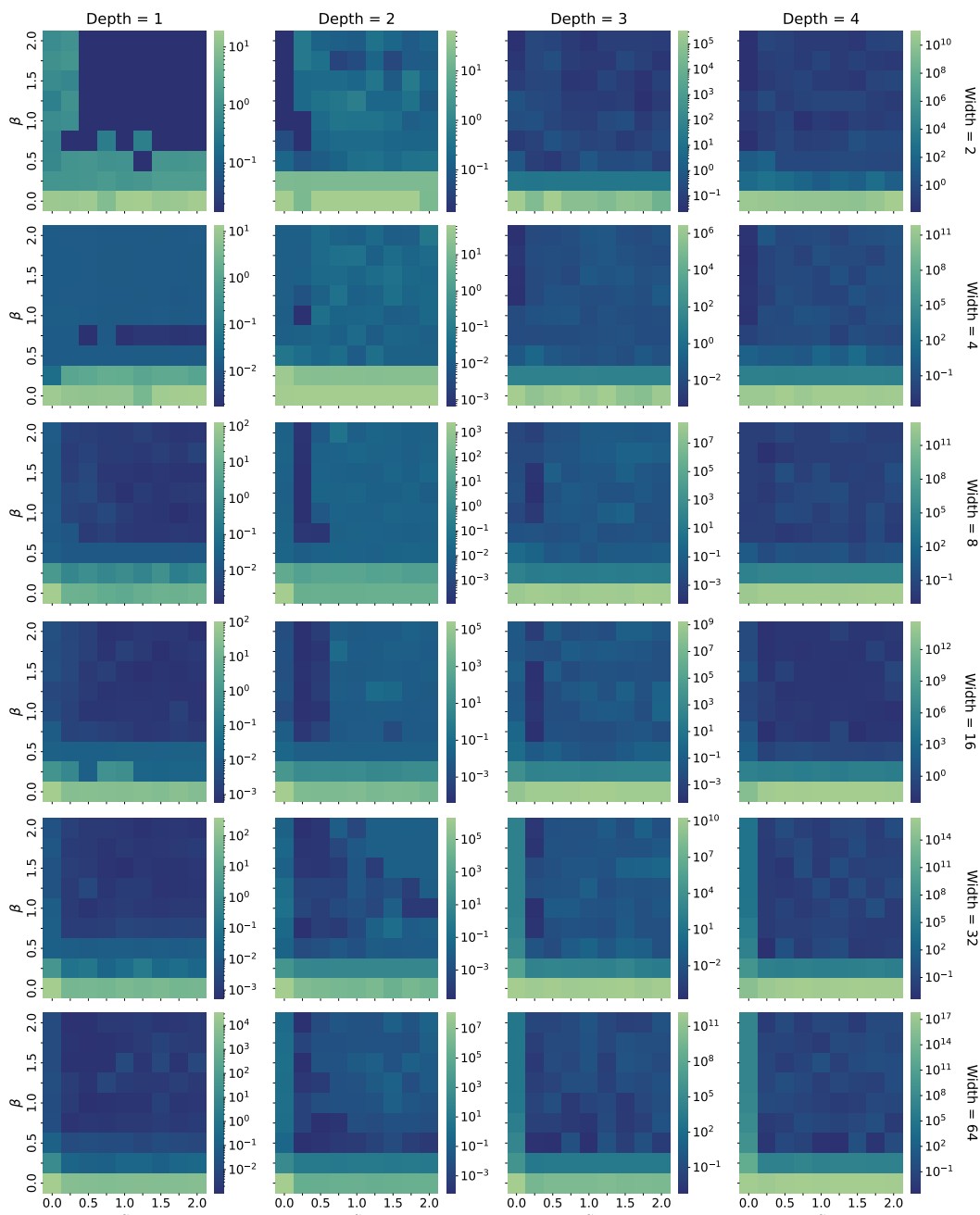

Figure 13: Grid search for the power-law initialization applied for the solution of the Allen–Cahn equation for $G = 20$. Each heatmap corresponds to an architecture, with the horizontal and vertical axis representing $\alpha$ and $\beta$, respectively, and color denoting final training loss.

# E  TRAINING CURVES WITH LEARNING RATE SCHEDULING

In Section 4.2 of the main text, the training curves shown in Figures 1 and 2 were obtained using a fixed learning rate in order to isolate the effect of initialization, as initialization and learning-rate adaptability are known to interact (e.g., (Yang et al., 2021)). However, the fixed learning rate induces oscillations in the loss curves, particularly for the Glorot and power-law schemes in the larger architectures. To verify that these oscillations are purely an artifact of the constant learning rate, we repeat the same training experiments using a learning-rate scheduler.

Figure 14 shows the results for the function fitting benchmarks, where all settings are identical to those in the main text except for the use of a learning-rate scheduler: training begins with a learning rate of $10^{-3}$, followed by exponential decay with decay factor $0.9$ every $50$ iterations. Similarly, Figure 15 shows the results for the PDE benchmarks, again using all the same hyperparameters as

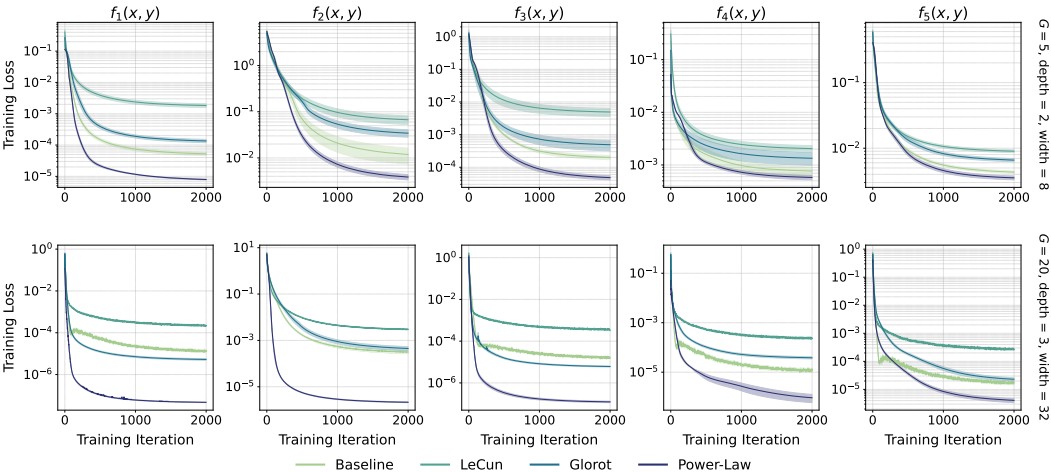

Figure 14: Training loss curves for function fitting benchmarks under baseline, LeCun-numerical, Glorot and power-law ($\alpha = 0.25, \beta = 1.75$) initializations when using a learning-rate scheduler. Results are averaged over five seeds, with shaded regions indicating the standard error. Top row: "small" architecture ($G = 5$, two hidden layers with 8 neurons each). Bottom row: "large" architecture ($G = 20$, three hidden layers with 32 neurons each).

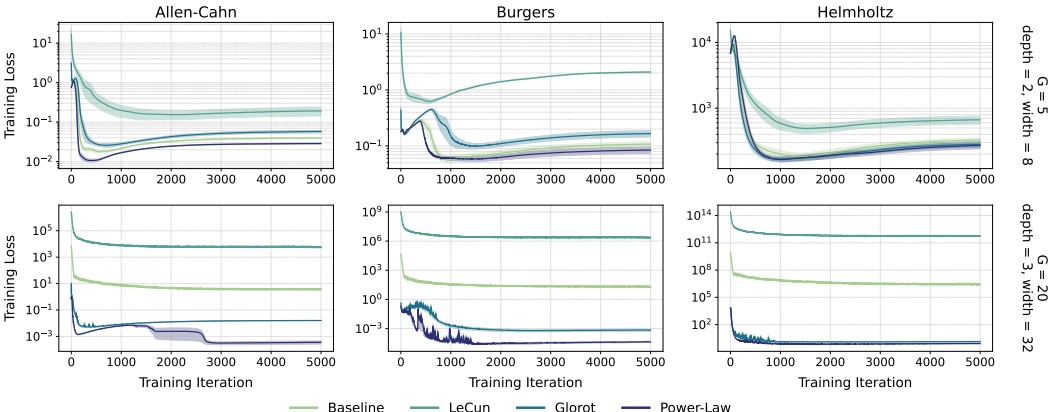

Figure 15: Training loss curves for forward PDE benchmarks under baseline, LeCun-numerical, Glorot and power-law ($\alpha = 0.25, \beta = 1.75$) initializations when using a learning-rate scheduler. Results are averaged over five seeds, with shaded regions indicating the standard error. Top row: "small" architecture ($G = 5$, two hidden layers with 8 neurons each). Bottom row: "large" architecture ($G = 20$, three hidden layers with 32 neurons each).

in the main text except for the scheduler: training begins with a learning rate of $10^{-3}$, followed by exponential decay with decay factor $0.85$ every $100$ iterations.

Across both sets of benchmarks, the learning-rate schedulers eliminate the oscillatory behavior observed under a fixed learning rate, yielding smoother training curves. Quantitatively, the final losses are slightly higher than those reported in the main text, due to the learning rate decaying even in regions where a larger step size would allow for further progress. Nonetheless, this influences only the numerical values: the qualitative picture remains unchanged, and the relative performance ordering of the initialization schemes is consistent with the fixed learning rate setting.

# F  NEURAL TANGENT KERNEL ANALYSIS

In this work, we use NTK analysis (Jacot et al., 2018) to better understand the effect of initialization schemes on function fitting and PDE benchmarks, both in terms of stability and conditioning.

## F.1  NTK FOR PIML WITH RBA WEIGHTS

In this subsection, we derive the NTK formalism used in our PDE experiments. Specifically, we extend the standard NTK framework for PIML (Wang et al., 2022) to cover the RBA-weighted loss function of Eq. (31).

We denote the PDE and boundary/initial condition residuals at the $i$-th collocation point by $r_i^{(\text{pde})}$ and $r_i^{(\text{bc})}$, respectively, as in Appendix C.2. We may re-weight the loss function of Eq. (31) to follow Wang et al. (2022) and subsequently write it in vector form as

$$\mathcal{L}(\boldsymbol{\theta}) \;=\; \frac{1}{2}\left\|\tilde{\mathbf{r}}^{(\text{pde})}(\boldsymbol{\theta})\right\|_2^2 \;+\; \frac{1}{2}\left\|\tilde{\mathbf{r}}^{(\text{bc})}(\boldsymbol{\theta})\right\|_2^2, \qquad \tilde{\mathbf{r}}^{(\xi)} = \mathbf{A}^{(\xi)}\mathbf{r}^{(\xi)}, \tag{33}$$

where $\mathbf{r}^{(\xi)}$ stacks the residuals of type $\xi \in \{\text{pde}, \text{bc}\}$, $\boldsymbol{\alpha}^{(\xi)} = (\alpha_1^{(\xi)}, \ldots, \alpha_{N_\xi}^{(\xi)})^\top$ are the RBA weights and $\mathbf{A}^{(\xi)} = \text{diag}(\boldsymbol{\alpha}^{(\xi)})$. Throughout a single gradient step we treat $\boldsymbol{\alpha}^{(\xi)}$ as constants, as they are updated only between steps by Eq. (32), outside of the gradient descent scheme.

Let $\mathbf{J}^{(\xi)}(\boldsymbol{\theta}) \in \mathbb{R}^{N_\xi \times P}$ be the Jacobian of the residuals with respect to the parameters, i.e., its $i$-th row is $\mathbf{J}_i^{(\xi)}(\boldsymbol{\theta}) = \partial r_i^{(\xi)}(\boldsymbol{\theta})/\partial \boldsymbol{\theta}^\top$. For a parameter update $\Delta\boldsymbol{\theta} = -\eta\,\nabla_{\boldsymbol{\theta}}\mathcal{L}(\boldsymbol{\theta})$, a first-order expansion around $\boldsymbol{\theta}$ yields,

$$\Delta\tilde{\mathbf{r}}^{(\xi)}(\boldsymbol{\theta}) \;=\; \mathbf{A}^{(\xi)}\Delta\mathbf{r}^{(\xi)}(\boldsymbol{\theta}) \;\approx\; \mathbf{A}^{(\xi)}\mathbf{J}^{(\xi)}(\boldsymbol{\theta})\,\Delta\boldsymbol{\theta}. \tag{34}$$

Using Eq. (33) and the chain rule, the full-batch gradient is

$$\nabla_{\boldsymbol{\theta}}\mathcal{L}(\boldsymbol{\theta}) \;=\; \sum_{i=1}^{N_{\text{pde}}} \tilde{r}_i^{(\text{pde})}(\boldsymbol{\theta})\,\nabla_{\boldsymbol{\theta}}\tilde{r}_i^{(\text{pde})}(\boldsymbol{\theta}) \;+\; \sum_{i=1}^{N_{\text{bc}}} \tilde{r}_i^{(\text{bc})}(\boldsymbol{\theta})\,\nabla_{\boldsymbol{\theta}}\tilde{r}_i^{(\text{bc})}(\boldsymbol{\theta}). \tag{35}$$

Since $\tilde{r}_i^{(\xi)} = \alpha_i^{(\xi)}\,r_i^{(\xi)}$ and $\boldsymbol{\alpha}^{(\xi)}$ is held fixed within the step,

$$\nabla_{\boldsymbol{\theta}}\tilde{r}_i^{(\xi)}(\boldsymbol{\theta}) \;=\; \alpha_i^{(\xi)}\nabla_{\boldsymbol{\theta}}r_i^{(\xi)}(\boldsymbol{\theta}) \;=\; \alpha_i^{(\xi)}\big(\mathbf{J}_i^{(\xi)}(\boldsymbol{\theta})\big)^\top. \tag{36}$$

Substituting $\Delta\boldsymbol{\theta} = -\eta\,\nabla_{\boldsymbol{\theta}}\mathcal{L}(\boldsymbol{\theta})$ into Eq. (34) and grouping terms gives the linear dynamics

$$\Delta\tilde{\mathbf{r}}^{(\xi)}(\boldsymbol{\theta}) \;\approx\; -\eta\Bigg[ \underbrace{\big(\mathbf{A}^{(\xi)}\mathbf{J}^{(\xi)}(\boldsymbol{\theta})\big)\big(\mathbf{A}^{(\text{pde})}\mathbf{J}^{(\text{pde})}(\boldsymbol{\theta})\big)^\top}_{\widetilde{\mathbf{K}}^{(\xi,\text{pde})}}\tilde{\mathbf{r}}^{(\text{pde})}(\boldsymbol{\theta})$$

$$+\; \underbrace{\big(\mathbf{A}^{(\xi)}\mathbf{J}^{(\xi)}(\boldsymbol{\theta})\big)\big(\mathbf{A}^{(\text{bc})}\mathbf{J}^{(\text{bc})}(\boldsymbol{\theta})\big)^\top}_{\widetilde{\mathbf{K}}^{(\xi,\text{bc})}}\tilde{\mathbf{r}}^{(\text{bc})}(\boldsymbol{\theta})\Bigg]. \tag{37}$$

As mentioned in the main text, Eq. (37) shows that the weighted residual vectors $\tilde{\mathbf{r}}^{(\xi)}$ evolve under a weighted NTK with blocks

$$\widetilde{\mathbf{K}}^{(\xi,\zeta)} \;=\; \big(\mathbf{A}^{(\xi)}\mathbf{J}^{(\xi)}\big)\big(\mathbf{A}^{(\zeta)}\mathbf{J}^{(\zeta)}\big)^\top, \qquad \xi, \zeta \in \{\text{pde}, \text{bc}\}. \tag{38}$$

## F.2 NTK Spectra for Varying Power-Law Exponents

To complement the heatmaps of Appendix D and to further illustrate the robustness of the power-law initialization, we examine in this Appendix how the NTK spectrum varies across different $(\alpha, \beta)$ configurations. The goal of this analysis is twofold. First, it provides an NTK-based view of the "good regions" identified in the grid search, showing how favorable exponent choices correspond to well-conditioned and stable kernels. Second, it demonstrates that the power-law scheme is not sensitive to a single finely tuned pair of exponents: more than one $(\alpha, \beta)$ combinations within the identified range yield well-behaved spectra. This supports the idea that one may tune the exponents once per problem domain and thereafter select any configuration from the favorable region.

Figure 16 displays the NTK eigenvalue spectra for all exponent pairs considered in the grid search, using the "large" architecture ($G = 20$, three hidden layers with 32 neurons each) and the function fitting target $f_3(x, y)$. Figure 17 shows the corresponding results for the PDE residual term of Burgers' equation. In both cases, well-conditioned spectra concentrate in the same regions suggested by the grid-search results.

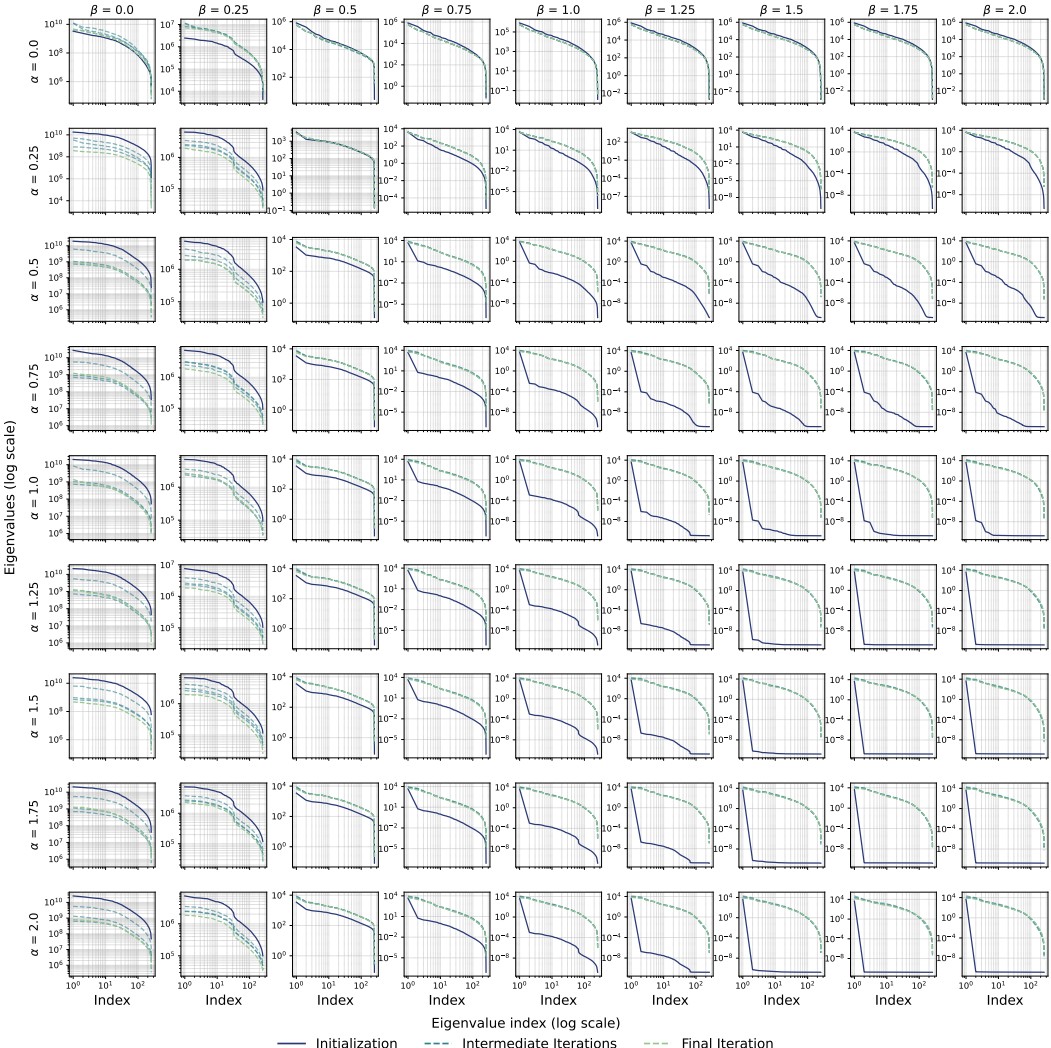

Figure 16: NTK eigenvalue spectra for the large architecture ($G = 20$, three hidden layers with 32 neurons each) on the function fitting target $f_3(x, y)$, shown for all $(\alpha, \beta)$ configurations considered in the grid search. Each panel corresponds to one exponent pair and displays spectra at initialization, mid-training, and convergence.

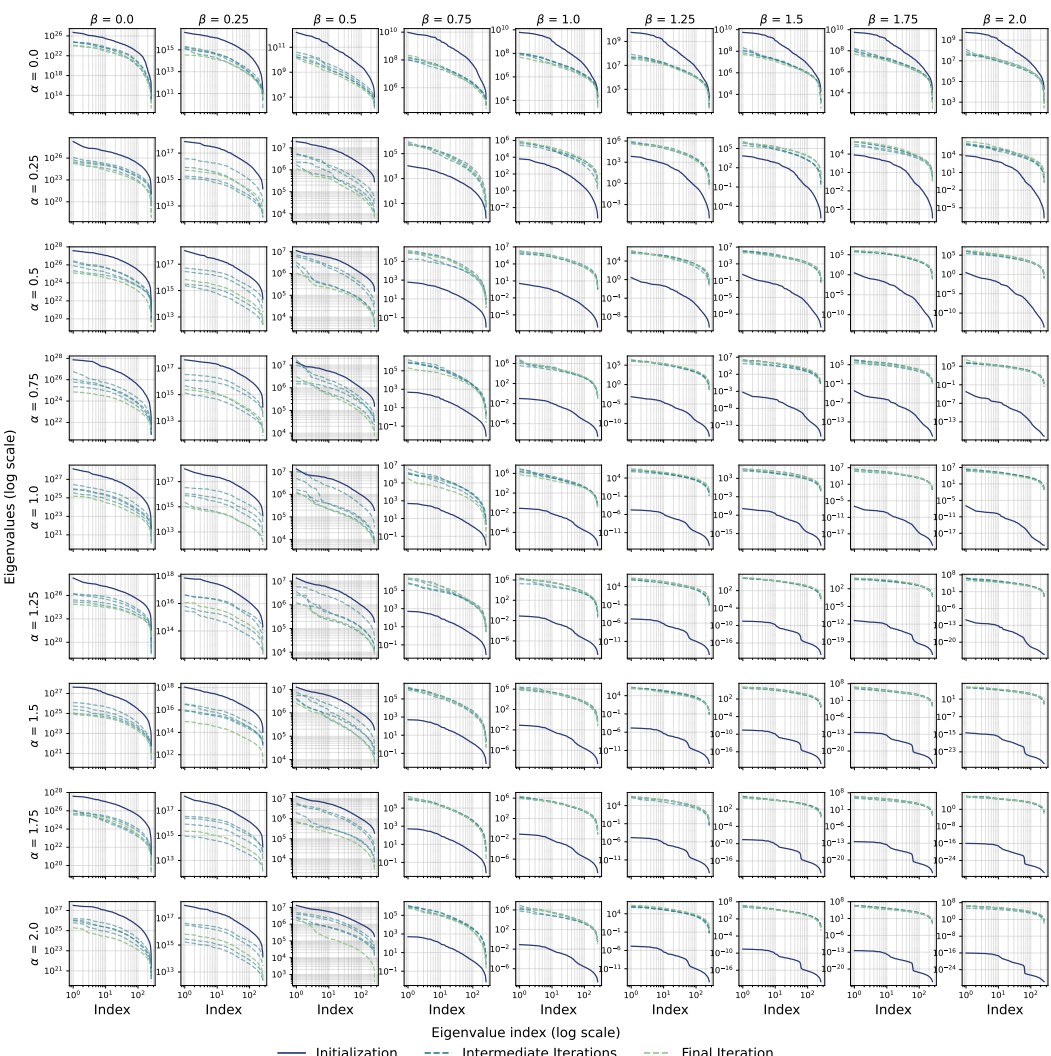

Figure 17: NTK eigenvalue spectra for the large architecture ($G = 20$, three hidden layers with 32 neurons each) on the PDE residual term of Burgers' equation, shown for all $(\alpha, \beta)$ configurations considered in the grid search. Each panel corresponds to one exponent pair and displays spectra at initialization, mid-training, and convergence.

### F.3 ADDITIONAL NTK SPECTRA

For completeness, we report additional NTK spectra not included in the main text. Figures 18–21 show the results for the remaining function fitting benchmarks ($f_1$, $f_2$, $f_4$, $f_5$), while Figures 22, 23 correspond to the Burgers' and Helmholtz PDEs. All results are obtained using the "large" architecture ($G = 20$, three hidden layers with 32 neurons each) and values $\alpha = 0.25$, $\beta = 1.75$ for the power-law initialization, consistent with the setting analyzed in Section 4.2.

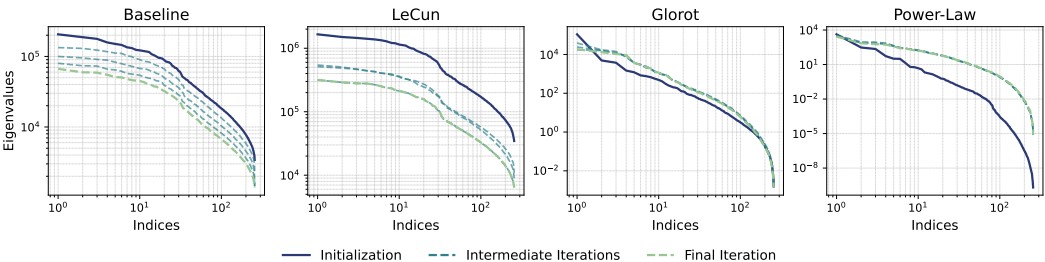

Figure 18: Eigenvalue spectra of the NTK matrix at initialization (solid blue), intermediate iterations (dashed teal), and final iteration (dashed green) for function fitting benchmark $f_1(x, y)$ under different initialization strategies. Results correspond to the "large" architecture ($G = 20$, three hidden layers with 32 neurons each). The power-law initialization uses $\alpha = 0.25, \beta = 1.75$.

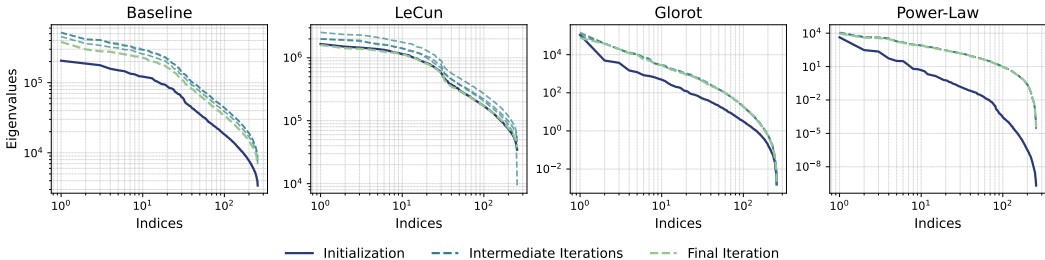

Figure 19: Eigenvalue spectra of the NTK matrix at initialization (solid blue), intermediate iterations (dashed teal), and final iteration (dashed green) for function fitting benchmark $f_2(x, y)$ under different initialization strategies. Results correspond to the "large" architecture ($G = 20$, three hidden layers with 32 neurons each). The power-law initialization uses $\alpha = 0.25, \beta = 1.75$.

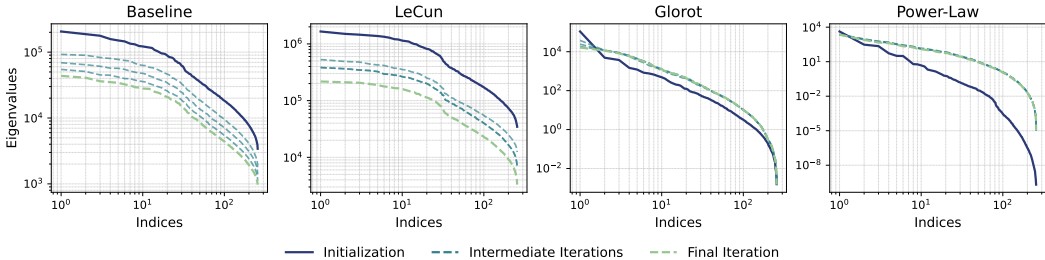

Figure 20: Eigenvalue spectra of the NTK matrix at initialization (solid blue), intermediate iterations (dashed teal), and final iteration (dashed green) for function fitting benchmark $f_4(x, y)$ under different initialization strategies. Results correspond to the "large" architecture ($G = 20$, three hidden layers with 32 neurons each). The power-law initialization uses $\alpha = 0.25, \beta = 1.75$.

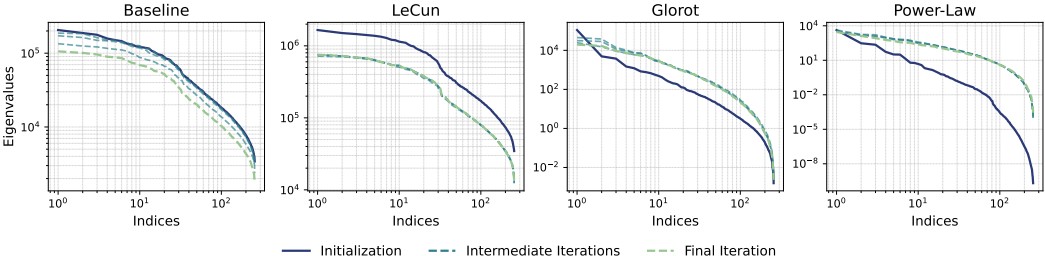

Figure 21: Eigenvalue spectra of the NTK matrix at initialization (solid blue), intermediate iterations (dashed teal), and final iteration (dashed green) for function fitting benchmark $f_5(x, y)$ under different initialization strategies. Results correspond to the "large" architecture ($G = 20$, three hidden layers with 32 neurons each). The power-law initialization uses $\alpha = 0.25, \beta = 1.75$.

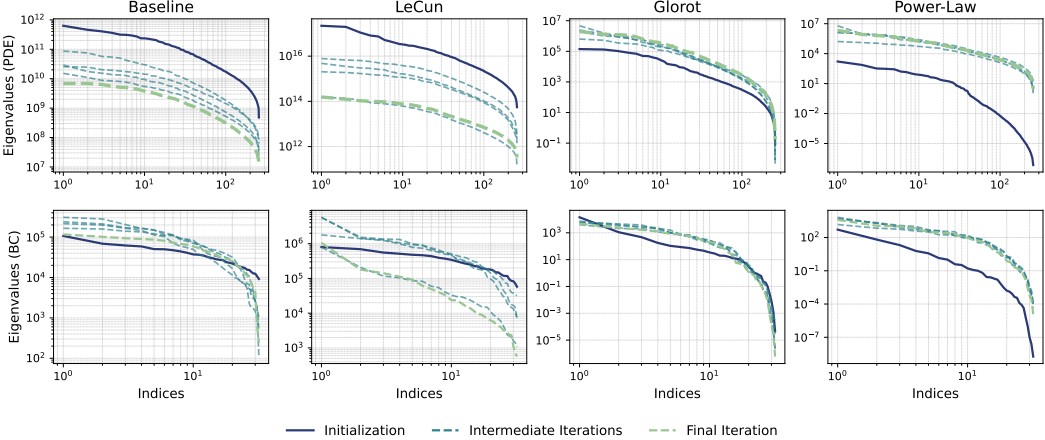

Figure 22: NTK eigenvalue spectra for the Burgers' PDE benchmark under baseline, LeCun-numerical, Glorot, and power-law ($\alpha = 0.25, \beta = 1.75$) initializations. Top row: spectra corresponding to the PDE residual term. Bottom row: spectra for the boundary/initial condition terms. Solid blue lines show the initialization, dashed teal lines show intermediate iterations, and dashed green lines show the final iteration.

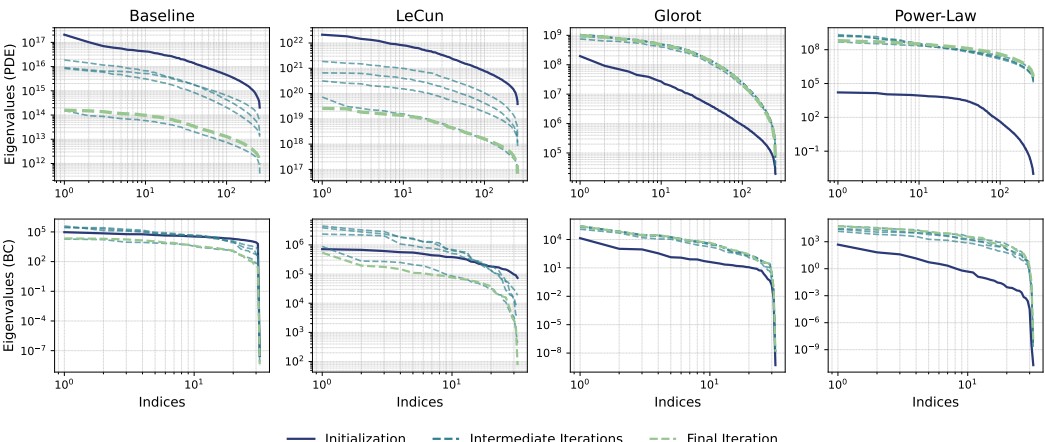

Figure 23: NTK eigenvalue spectra for the Helmholtz PDE benchmark under baseline, LeCun-numerical, Glorot, and power-law ($\alpha = 0.25, \beta = 1.75$) initializations. Top row: spectra corresponding to the PDE residual term. Bottom row: spectra for the boundary/initial condition terms. Solid blue lines show the initialization, dashed teal lines show intermediate iterations, and dashed green lines show the final iteration.

