# OpenReview forum: "Initialization Schemes for Kolmogorov–Arnold Networks: An Empirical Study"
_ICLR.cc/2026/Conference — ICLR 2026 Poster_

### Official Review · Reviewer_c2rF · 2025-10-27

**Soundness:** 2
**Presentation:** 3
**Contribution:** 3
**Rating:** 6
**Confidence:** 2

**Summary:**

This paper investigates initialization strategies for spline-based Kolmogorov-Arnold Networks (KANs), a relatively new neural network architecture. The authors argue that proper initialization is crucial for KAN performance but remains under-explored, and methods from traditional MLPs are not directly applicable. The paper proposes and evaluates several initialization schemes: two theoretically-motivated approaches inspired by LeCun and Glorot/Xavier variance preservation principles (including a variant using batch-normalized spline bases), and an empirical power-law family with tunable exponents. These schemes are evaluated through large-scale grid searches on function fitting and forward PDE approximation tasks, analysis of training dynamics (loss curves, Neural Tangent Kernel spectrum), and testing on a subset of the Feynman dataset. Key findings suggest that the Glorot-inspired scheme performs well, especially for larger models, while the empirical power-law initialization consistently achieves the best performance across tasks and architectures.

**Strengths:**

1: Addresses a critical and timely aspect of training KANs – initialization,  and provides the first systematic comparison of different initialization strategies specifically tailored for spline-based KANs.

2: The proposed power-law initialization demonstrably leads to superior performance across a wide range of settings in the experiments, offering a practical and effective strategy. The finding that small $\alpha$ and large $\beta$ are generally preferred provides a useful heuristic.

**Weaknesses:**

1. The paper is purely empirical, lacking a theoretical justification for the effectiveness of methods. For example, why do small $\alpha$ and large $\beta$ work well for power-law initialization? If relying solely on empirical search, there may be a suspicion of parametric-innovation.

2. The data in experimental is relatively simple, considering only function fitting and PDE, which limits the generalizability of the author's conclusions.

3. Lack of in-depth analysis on the reasons for poor initialization performance, such as for LeCun-normalized.

**Questions:**

1. Could you comment on how the performance of KANs with the proposed initializations compares to standard, well-initialized MLPs on some of the benchmark tasks?

2.Could the sensitivity to the variance equipartition assumption in the LeCun/Glorot derivations be investigated? What happens if different partitions are assumed?

3.Can the authors provide more insight into why the LeCun-normalized scheme failed to perform well? Is there an interaction between batch normalization and the spline basis learning?

---

> ### Author Response · Authors · 2025-11-17
> **Response to reviewer c2rF (Part 1)**
>
> We thank the reviewer for the constructive and thoughtful feedback. Several of the points raised helped us improve the clarity and depth of the revised manuscript. We address each item below.
>
>
> ### Response to Weakness 1
>
> We fully agree that the power-law initialization, as proposed, is empirical in nature. This is precisely why the paper includes a detailed NTK-based analysis, which provides insight beyond the grid-search outcomes. We also note that the manuscript introduces two theoretically motivated schemes, the LeCun- and Glorot-inspired initializations, with the latter emerging as a strong, principled alternative to the baseline across architectures and tasks.
>
> To strengthen the empirical justification for the power-law family, the revised manuscript now includes a new appendix (Appendix F.2) containing NTK spectra for all $(\alpha, \beta)$ configurations for both a representative function fitting benchmark ($f_3$) and a PDE benchmark (Burgers). These plots highlight how kernel conditioning varies across exponent choices and corroborate the regions identified by the grid searches.
>
> Importantly, the results indicate that the power-law method is robust, with favorable conditioning and performance arising not from a single finely tuned pair of exponents, but from broader regions of exponent values. To further underline this, in Section 4.2 we intentionally chose a $\beta$ value outside the optimal range identified for PDEs and still observed performance superior to Glorot and the baseline. We now make this point explicit in the text (see lines 334–337).
>
>
> ### Response to Weakness 2
>
> We acknowledge the reviewer's point, which is mentioned in Section 5.1 in the limitations of our work. Our goal was to evaluate initialization schemes in the domains where spline-based KANs have seen their most widespread use to date, namely scientific applications involving regression and physics-informed machine learning. These are precisely the contexts where KANs have demonstrated competitive or state-of-the-art performance (see [1-3]). To this day, KANs have not yet shown impactful results in domains such as vision or language modeling, and this can be mainly attributed to the fact that they were not designed for these domains [4].
>
>
> ### Response to Weakness 3
>
> We appreciate this concern and have substantially expanded the analysis related to LeCun initialization in the revised manuscript. In particular:
>
> * We now explain why the LeCun-normalized variant performs better than its numerical counterpart (see lines 225–227) in function fitting. Additionally, we provide new text explaining why the normalized variant fails completely in PDE settings (the normalization propagates through derivatives, distorting the relative scaling between function values and derivatives involved in the PDE residuals - see lines 269-273).
>
> * Unlike the original submission, we now include the numerical LeCun variant in all experiments of Section 4.2, enabling direct comparison. This also allowed us to add commentary on its loss curves and NTK spectra, which provide additional insight into its poor performance.
>
>
> ### Reply to Question 1
>
> The scope of the paper is to compare initialization schemes for KANs, not to compare KANs to other architectures. Nevertheless, the original KAN paper [2] evaluates KANs against parameter-matched MLPs on the a subset of the Feynman dataset as the one used in our Section 4.3. Their results show that KANs perform as well as or better than MLPs in the majority of cases. Since those results were obtained using the baseline KAN initialization, it is reasonable to deduce that our improved Glorot and power-law initializations would yield even better performance relative to MLPs.
>
>
> ### Reply to Question 2
>
> We indeed investigated an alternative variance split—assigning equal variance to residual and basis terms and we retain this comment in the revised manuscript. The performance of this alternative split was consistently poorer. A 50–50 partition can be justified for non-fully-adaptive basis functions (here we use the terminology of [5]), which may lack local support across the entire input domain. In such cases, the residual term compensates for missing expressivity. However, spline bases (as used in KANs) are fully adaptive and provide local support across the entire input domain, making the equal split less meaningful. For this reason, the equipartition used in the derivation was the most justified choice.
>
>
> *Please see part 2 for the rest of the response*

---

> > ### Author Response · Authors · 2025-11-17
> > **Response to reviewer c2rF (Part 2)**
> >
> > *This is the continuation of our first response*
> >
> > ### Reply to Question 3
> >
> > We especially thank the reviewer for raising this point, as this feedback helped us significantly improve the clarity of the manuscript. In the original draft, we used the phrase "batch-normalized basis functions", which was potentially misleading because this has nothing to do with batch normalization in the neural network sense. We have revised the terminology to "normalized basis functions", which simply refers to standardizing each basis function so that its outputs have unit variance (see Eq. (4)).
> >
> > As discussed above, the reviewer's intuition into this is correct: the LeCun-normalized scheme performs poorly for PDEs precisely because this normalization propagates undesirably to the derivatives of the basis functions, which appear in the PDE residual. The revised manuscript now makes this point explicit.
> >
> >
> > ### References:
> >
> > [1] Noorizadegan, A., Wang, S., & Ling, L. (2025). A practitioner’s guide to Kolmogorov-Arnold networks (No. arXiv:2510.25781). arXiv.
> >
> > [2] Liu, Z., Wang, Y., Vaidya, S., Ruehle, F., Halverson, J., Soljacic, M., Hou, T. Y., & Tegmark, M. (2024, October 4). Kan: Kolmogorov–Arnold networks. The Thirteenth International Conference on Learning Representations.
> >
> > [3] Shukla, K., Toscano, J. D., Wang, Z., Zou, Z., & Karniadakis, G. E. (2024). A comprehensive and FAIR comparison between MLP and KAN representations for differential equations and operator networks. Computer Methods in Applied Mechanics and Engineering, 431, 117290.
> >
> > [4] Liu, Z., Ma, P., Wang, Y., Matusik, W., & Tegmark, M. (2024). Kan 2. 0: Kolmogorov-arnold networks meet science (No. arXiv:2408.10205). arXiv.
> >
> > [5] Rigas, S., Papachristou, M., Papadopoulos, T., Anagnostopoulos, F.K., & Alexandridis, G. (2024). Adaptive Training of Grid-Dependent Physics-Informed Kolmogorov-Arnold Networks. IEEE Access, 12, 176982-176998.

---

> > > ### Comment · Reviewer_c2rF · 2025-11-21
> > >
> > > The author acknowledged weakness and answered my question, but since I am not fully confident in this field, I have decided to maintain this score.

---

> > > > ### Author Response · Authors · 2025-11-21
> > > >
> > > > Once again, thank you for your review and for helping improve the manuscript! We remain at your disposal throughout the remainder of the discussion period for anything else.

---

### Official Review · Reviewer_tHHq · 2025-10-29

**Soundness:** 2
**Presentation:** 3
**Contribution:** 2
**Rating:** 4
**Confidence:** 3

**Summary:**

The paper explores three new initialization schemes for KANs, identifying a current gap that KAN implementations rely on the standard initialization presented in the original KAN paper. The first two initialization methods explored LeCun and Glorot, are based in theory, the first driven by variance preservation, in which the variance of the inputs should match the variance of the outputs (LeCun) and the other driven by stabilizing variance for activations and gradients across layers. Additionally, the paper explores a third, empirical initialization based on the power-law scaling of KAN structure, that includes tunable exponents on the KAN's architecture (termed "power-law" in the paper). The paper goes on to evaluate the accuracy of these initialization methods on two main tasks, function fitting on five 2D target functions, and forward PDE problems. On both of these tasks, power-law initialization significantly outperforms other methods, with Glorot initialization as a distant second method. Additionally, the authors explore training stability, comparing loss curves and eigenvalue spectra of the NTK matrix.

**Strengths:**

The paper addresses an open question in KAN training and successfully demonstrates different initialization variants and their relative benefits. The paper motivates well the use of the two theoretically-driven initialization methods, and selects a good set of tasks and demonstrations to illustrate the relative benefits of each method. This is very fundamental work that is important to enabling the community to practically use KANs.

**Weaknesses:**

I appreciated the theoretical motivation of LeCun and Glorot initializations but was disappointed that there wasn't much discussions on why these methods behaved in the way they did. Table 1 for example shows that only about 15% of runs with LeCun initialization outperform the baseline, but there isn't an exploration into why this might be the case, or an exploration of the dynamics of this method of initialization. The same holds true for the results in Table 2.

I'm not quite sure I understand what is being demonstrated in Figure 1 for the large architecture, as training instability for the power law initialization dominates the dynamics. The authors say "The more pronounced oscillations observed in its curves stem from the fact that the model already reaches very low loss values under a fixed learning rate, leaving no smoothing effect from adaptive adjustments." but the learning rate in these plots is fixed to 10e-3 so I'm not sure what is meant by "no smoothing effect from adaptive adjustments" unless the learning rate is in fact changing. The same instability concern holds true for the PDE case.

Overall, I have concerns with the power law initialization, as I believe this method is prone to overfitting, and that results of this method are artificially inflated. I'm not quite sure that comparing the performance of this method with the theoretically motivated initializations.

**Questions:**

I would love to understand better what is driving the dynamics of the LeCun and Glorot initialization methods. I would also like to better understand the decision to add tunable parameters to an initialization method, whether there is a precedent here, and why we believe that would not result in overfitting.

---

> ### Author Response · Authors · 2025-11-17
> **Response to reviewer tHHq (Part 1)**
>
> We thank the reviewer for the thoughtful and constructive feedback. All of the concerns raised have helped us significantly improve the clarity and quality of the manuscript. Below we address each point in turn.
>
> ### Response to Weakness 1
>
> We agree that the discussion in the original submission was limited for these initializations. With the additional space permitted during the rebuttal period, we have rewritten the commentary throughout Section 4 to give a more detailed explanation, especially for LeCun, as Glorot does much better than the baseline overall and is a valid theory-driven alternative. In particular:
>
> * We now explicitly discuss why the LeCun-normalized variant performs better than the numerical version in function fitting: the normalization enforces unit-variance outputs per basis element, which aligns perfectly with the variance-preservation objective (see lines 225–227).
>
> * We also explain why the normalized LeCun variant fails in PDE settings: the normalization of the spline basis functions propagates to derivatives, which directly enter the PDE residuals. This distorts the scaling of derivatives relative to function values (see lines 269–273).
>
> * We now include LeCun (numerical) in all experiments of Section 4.2 instead of discarding it. This allowed us to add commentary on its loss curves as well as NTK spectra, which provide further insight into why the method underperforms.
>
>
> ### Response to Weakness 2
>
> We appreciate the reviewer pointing this out. The original sentence was poorly phrased, and we have now rewritten this section for clarity. The reason why the oscillations appear is that we (intentionally) did not apply learning rate scheduling. Without scheduling, once the loss becomes very small, a fixed step size produces sharp updates and visible oscillations; this is a well-known effect in optimization with fixed learning rates (and also the original motivation for the use of scheduling). This is now clearly stated in the revised text (see lines 327–333 and 343–345).
>
> We avoided learning rate scheduling because initialization and learning rate scheduling are known to interact (see e.g., [1]). Adding a scheduler would introduce another confounding factor in the analysis of initialization schemes. Nonetheless, to demonstrate that the oscillations are simply a consequence of the fixed learning rate and not training instability, we added a new Appendix (Appendix E). There, we repeat the same experiments with learning rate scheduling. The resulting loss curves are substantially smoother, yet the qualitative ordering remains unchanged, confirming that the oscillations are not indicative of problematic optimization dynamics.
>
>
> ### Response to Weakness 3
>
> We understand the concern, and we agree that the original text did not clearly communicate the intended usage of the power-law initialization. Our aim is not to tune $(\alpha, \beta)$ for every new task. Instead, the idea is to tune the exponents once per problem domain (e.g., function fitting, PDE solving) and then reuse the resulting ranges. We have clarified this point in Section 3.1 (see lines 172–175).
>
> The manuscript already reflects this philosophy:
>
> * In Section 4.3 (Feynman dataset), we use a fixed single $(\alpha, \beta)$ pair without any additional tuning (see lines 442-443), and still achieve strong improvements compared to the other initialization methods. Interestingly, the functions studied have completely different properties (most importantly, different dimensionalities) compared to the ones that were used for the "tuning" process (the grid search of Section 4.1).
>
> * In the PDE experiments of Section 4.2, we intentionally choose a $\beta$ value outside the optimal region identified for PDEs. Despite this, the power-law scheme still outperforms all alternatives. We believe that this further demonstrates the robustness of the method (see lines 335–337).
>
> * All figures from Section 4.2 onwards (including Appendices) have been updated to explicitly state that the fixed pair $(\alpha=0.25, \beta=1.75)$ is used for all subsequent experiments.
>
> To further address concerns about overfitting, we have added a new appendix (Appendix F.2), where we provide NTK spectra across the full $(\alpha, \beta)$ grid for representative function-fitting ($f_3$) and PDE (Burgers) benchmarks. These results show that favorable conditioning occurs across regions of exponent values and not at isolated, finely tuned points.
>
> Finally, Section 4.1 now explicitly reports the performance of a single fixed $(\alpha, \beta)$ configuration (see lines 238–242 and 282–284), showing that a single fixed pair of $(\alpha, \beta)$ can outperform (by a wide margin) the other initialization methods, without comparing only the best pair per run to the others.
>
>
> *Please see part 2 for the rest of the response*

---

> > ### Author Response · Authors · 2025-11-17
> > **Response to reviewer tHHq (Part 2)**
> >
> > *This is the continuation of our first response*
> >
> > ### Reply to Question
> >
> > Based on the actions we took to strengthen the weaknesses identified above, we believe that we have also addressed the reviewer's question regarding the use of tunable exponents and the drivers for the dynamics of LeCun and Glorot (especially through the lens of the NTK spectra).
> >
> >
> > ### References:
> >
> > [1] Yang, G., Hu, E., Babuschkin, I., Sidor, S., Liu, X., Farhi, D., Ryder, N., Pachocki, J., Chen, W., & Gao, J. (2021). Tuning large neural networks via zero-shot hyperparameter transfer. Advances in Neural Information Processing Systems, 34, 17084–17097.

---

### Official Review · Reviewer_bFkh · 2025-10-31

**Soundness:** 2
**Presentation:** 3
**Contribution:** 3
**Rating:** 6
**Confidence:** 3

**Summary:**

The paper proposes three families of initialization for spline‑based KANs: (i) LeCun‑inspired schemes (one ``numerical,'' one using batch‑normalized spline bases), (ii) a Glorot‑inspired scheme that balances forward/backward variances, and (iii) an empirical power‑law that scales residual/basis weights via exponents $\alpha, \beta$. Experiments include 2‑D function fitting, three forward PDEs (Allen–Cahn, Burgers, 2‑D Helmholtz), and a subset of the Feynman dataset; training‑dynamics are visualized with both loss curve analysis and NTK spectra. The numerics show that both Glorot and tuned power‑law tends to beat the original KAN baseline, and tuned power‑law is the strongest overall.

**Strengths:**

1. Overall the paper is clearly written with clear problem statement and clean formulation for the KAN layer. The LeCun/Glorot initializations are derived explicitly.
2. The paper include broad and thorough experimentation. The NTK spectra are a nice lens to visualize conditioning differences for different initialization.
3. Even though the paper is empirical, better initialization recipes in different scenarios for KANs are of practical relevance.

**Weaknesses:**

1. Although the authors claim that the power-law initialization has the best overall performance, its performance is based on a grid search over $\alpha,\beta$ and only the best‑performing configuration is retained. In practice, such tuning can be costly, and the paper offers no low‑budget recipe to make the scheme usable out of the box.
2. The LeCun/Glorot initialization are derived by applying the standard variance‑preserving idea to the KAN layer and rely on a strong equipartition assumption across the
$G+k+1$ terms. The paper itself notes this simplification and that an alternative split performed worse. Meanwhile, the best results come from the empirical power‑law, for which the paper offers no mechanistic explanation beyond trends in heatmaps. The authors acknowledge this limitation, but as it stands, the conceptual contribution feels incremental and the empirical win might due to a tuned heuristic.

**Questions:**

The author's best region favors small $\alpha$ and larger $\beta$. Could the authors provide a heuristic/NTK‑based rationale and show how eigenvalue conditioning varies across $(\alpha,\beta)$?

---

> ### Author Response · Authors · 2025-11-17
> **Response to reviewer bFkh**
>
> We thank the reviewer for the thoughtful and constructive feedback. All of the concerns raised have helped us significantly improve the quality of the manuscript. Below we address each point in turn.
>
> ### Response to Weakness 1
>
> We agree that performing a full grid search would be expensive in practical applications. To clarify this, the revised manuscript now emphasizes that the power-law exponents are not meant to be re-tuned for every individual task. Rather, the idea is to tune the exponents once per problem domain (e.g., function fitting or forward-PDE solving) and then reuse the identified ranges. This is now stated explicitly in the Introduction (see lines 59–63) and again in Section 3.1 (see lines 172–175).
>
> We also note that our experiments already follow this "tune once - use broadly" principle:
>
> * After the grid search in Section 4.1, we fix a single exponent pair and apply it successfully to function-fitting tasks of different dimensionality in Section 4.3, without additional tuning (see lines 442-443).
>
> * For PDEs (Section 4.2), we deliberately chose a $\beta$ value outside the most favorable region identified for PDEs (see lines 334-337). Even so, the power-law initialization remained superior in training loss, convergence and NTK conditioning. This robustness indicates that the method is not overly sensitive to the exact exponent choice.
>
>
> ### Response to Weakness 2
>
> The reviewer is correct that our LeCun/Glorot derivations assume an equal variance split among the $G+k+1$ spline and residual components. We explored the most natural alternative, i.e., splitting variance equally between residual and basis terms, which performed noticeably worse. Following the terminology introduced in [1], spline bases form a fully adaptive basis, thus providing local support everywhere. Therefore, while a partition like 50%/50% for residual terms/spline terms, respectively, would make sense for a set of basis functions which do not provide local support in all of the input regions, this is not the case for splines.
>
> Regarding the empirical nature of the power-law scheme: we agree. This is why we explicitly stress its empirical character throughout the paper and discuss this in the limitations. However, this is precisely why the paper includes a detailed NTK-based analysis to investigate why Glorot and power-law behave well while LeCun and the baseline often do not. To provide additional insight, the revised manuscript includes expanded commentary on the NTK spectra (see Section 4.2), explaining why Glorot and power-law exhibit favorable conditioning while LeCun and the baseline often do not. These additions were motivated both by reviewer suggestions and the additional space available during the rebuttal phase.
>
>
> ### Reply to Question
>
> We believe that the clarifications made in response to Weakness 1 also partially address this question. Our results indicate that the power-law scheme does not depend on a narrowly tuned choice of $(\alpha, \beta)$. Instead, it works reliably across ranges of values identified in the grid searches. In addition, following the reviewer's suggestion, we have added a new appendix (Appendix F.2) in which we present NTK spectra for all $(\alpha, \beta)$ configurations for the representative cases of $f_3$ and the Burgers equation. These results directly visualize how kernel conditioning varies across exponent choices and reinforce the identification of the favorable regions suggested by the grid-search heatmaps.
>
>
> ### References:
>
> [1] Rigas, S., Papachristou, M., Papadopoulos, T., Anagnostopoulos, F.K., & Alexandridis, G. (2024). Adaptive Training of Grid-Dependent Physics-Informed Kolmogorov-Arnold Networks. IEEE Access, 12, 176982-176998.

---

### Author Response · Authors · 2025-11-28
**Global Rebuttal Summary**

At the time the OpenReview leak was announced, only Reviewer c2rF had seen our rebuttal and replied that they were satisfied with our changes. Because further discussion is now frozen, we provide this concise summary of the revisions made during the rebuttal phase.

# Overview of the paper

This work presents the first systematic study of initialization strategies for Kolmogorov-Arnold Networks. We introduce two theory-driven schemes (LeCun-inspired and Glorot-inspired) and an empirical power-law family with exponents $\alpha, \beta$. We evaluate them through large grid searches, training-dynamics analysis (loss + NTK spectra) and experiments on function fitting, forward PDEs and a subset of the Feynman dataset.

# Summary of Key Improvements in the Rebuttal Revision

### 1. Clarified the practical use and robustness of the power-law initialization

All three reviewers raised concerns about tuning cost or potential overfitting. We substantially clarified the intended usage:

- **Exponents are tuned once per problem domain** (e.g., function fitting, PDEs), not per task. The introduction and Section 3.1 were updated to state this explicitly.

- The manuscript already follows this principle. In Section 4.3 (Feynman), we use **one fixed pair** $(\alpha,\beta)$ for functions of different dimensionality - no re-tuning occurs. In Section 4.2 (PDEs), we deliberately use a $\beta$ **outside the optimal region**, yet still outperform Glorot and the baseline, demonstrating robustness.

- Section 4.1 now reports performance of **one fixed pair** $(\alpha,\beta)$, not only the best pair, showing consistent superiority over all theory-driven schemes.

### 2. Added a new appendix with full NTK spectra across the entire $(\alpha,\beta)$ grid

To provide deeper insight about why certain exponent ranges work well, we added **Appendix F.2**, which presents NTK eigenvalue spectra for **all** $(\alpha,\beta)$ combinations, both for a representative function-fitting task ($f_3$) and for a PDE benchmark (Burgers). This analysis shows that well-conditioned spectra arise across broad regions of exponent values rather than at isolated points. This not only explains the robustness of the power-law method to the exact choice of $(\alpha,\beta)$, but also reinforces the grid-search observations with kernel-level evidence.

### 3. Expanded discussion on LeCun and Glorot performance

We revised Section 4 extensively to clarify the mechanisms behind each initialization:

- We now emphasize that **Glorot succeeds compared to the default initialization** and is a strong theory-driven alternative to the baseline.

- We clarified why LeCun-normalized performs better in function fitting than LeCun-numerical and also why it completely breaks down for PDEs.

- We restored LeCun (numerical) in all experiments following Section 4.1 and added commentary and NTK spectra showing why it underperforms compared to the other initialization schemes.

### 4. Clarified the cause of oscillations in loss curves and added scheduler-based experiments

Based on a comment about the oscillations in Figures 1-2, we rewrote the explanation entirely. Oscillations arise because **no learning-rate scheduling is applied** and a fixed learning rate can produce sharp updates once losses become small. We avoided scheduling intentionally to prevent confounding interactions with initialization. To confirm that oscillations were not instability, we added **Appendix E**, where all experiments are repeated *with* scheduling.

### 5. General improvements

- Many commentary sections in Section 4 were rewritten for clarity and depth.

- All figures from Section 4.2 onward were updated to explicitly state the fixed $(\alpha,\beta)$ values used.

- Terminology for "batch-normalized basis functions" was corrected to "normalized basis functions" to avoid confusion with batch normalization.

- The manuscript’s overall readability and coherence were significantly improved.

# Final Remarks

Reviewers raised important points that substantially improved the clarity and quality of the paper. Across all reviews, the main concerns involved explanation depth, robustness of the empirical method and clarity of certain results. The revised manuscript includes new appendices, new experiments, further NTK analyses, clearer explanations of failure modes and an explicit description of how the power-law method is meant to be used in practice. We hope this summary helps the new Area Chair in assessing the updated submission and we thank all reviewers for their constructive input.

---

### Meta-Review · Area_Chair_5SCS · 2026-01-05

**Summary:**

The reviewers broadly agreed that the paper addresses a timely, practically critical gap in Kolmogorov–Arnold Networks (KANs), as initialization can materially affect training outcomes. The reviewers are also raising concerns that make the recommendation borderline.
The key concerns include: (i) Practicality and fairness of the empirical power-law initialization. (ii) Insufficient mechanistic and theoretical explanation. (iii) Optimization dynamics and instability in loss curves. (iv) Limited scope and generalizability.
Given the initial scores (6/6/4) and the clarifications during rebuttal, I would lean weak accept, primarily based on the provided empirical gains for KANs and the strengthened analysis + framing in the revised version, while noting that some fundamental limitations remain.

**Reviewer Concerns:**

Concerns addressed in the rebuttal.

1.	Grid-search tuning cost and best-pair cherry-picking for power-law.

2.	Need for more mechanistic evidence for why these exponent regions work.

3.	Clarifying LeCun variants and the PDE failure mode.

4.	Some terminology confusion.

5.	Oscillatory loss curves and learning-rate schedule concerns.

Concerns still outstanding.

1.	Lack of a rigorous theory for the power-law scheme and principled exponent selection. Even after adding NTK grid evidence, the proposed method remains empirical. Thus, the reviewers’ concern beyond kernel-conditioning correlations remains unresolved. The authors explicitly acknowledged this as future work in the manuscript.

2.	Breadth of evaluation and comparative context vs MLPs. The paper still does not present a direct, controlled comparison to well-initialized MLPs on the included benchmarks. Instead, it points to the original KAN paper’s empirical comparisons.

3.	The empirical part in the revised manuscript does not fully satisfy the broader generalizability concern, e.g., other domains, other KAN variants, or interactions with adaptive optimizers.  The empirical results mainly focus on scientific regression and PDE contexts for spline-based KANs.

**Reviewer Scores:**

Reviewer bFkh: stays 6

Reviewer tHHq: from 4 to 6, possibly, as the authors provided empirical validations.

Reviewer c2rF: stays 6

---

### Decision · Program_Chairs · 2026-01-26

Accept (Poster)